# Distinct activation mechanisms of CXCR4 and ACKR3 revealed by single-molecule analysis of their conformational landscapes

Christopher T Schafer[1†], Raymond F Pauszek III[2‡], Martin Gustavsson[1§], Tracy M Handel[1]*, David P Millar[2]*

[1]Skaggs School of Pharmacy and Pharmaceutical Sciences, Department of Pharmacology, University of California San Diego, La Jolla, United States; [2]Department of Integrative Structural and Computational Biology, The Scripps Research Institute, La Jolla, United States

*For correspondence:
thandel@ucsd.edu (TMH);
millar@scripps.edu (DPM)

Present address: [†]Amsterdam Institute for Molecular and Life Sciences, Department of Medicinal Chemistry, Faculty of Science, Vrije Universiteit Amsterdam, Amsterdam, Netherlands; [‡]LUMICKS, Amsterdam, Netherlands; [§]Department of Biomedical Sciences, Faculty of Health and Medical Sciences, University of Copenhagen, Copenhagen, Denmark

## eLife Assessment

This manuscript describes the characterization of the conformational dynamics of two chemokine receptors at the single-molecule level using FRET. The authors make a **convincing** case for attributing the distinct interaction and pharmacology of the two receptors to differences in their conformational energy landscape. These **important** findings will be of interest to scientists working on activation mechanisms of GPCRs and signal transduction.

**Abstract** The canonical chemokine receptor CXCR4 and atypical receptor ACKR3 both respond to CXCL12 but induce different effector responses to regulate cell migration. While CXCR4 couples to G proteins and directly promotes cell migration, ACKR3 is G-protein-independent and scavenges CXCL12 to regulate extracellular chemokine levels and maintain CXCR4 responsiveness, thereby indirectly influencing migration. The receptors also have distinct activation requirements. CXCR4 only responds to wild-type CXCL12 and is sensitive to mutation of the chemokine. By contrast, ACKR3 recruits GPCR kinases (GRKs) and β-arrestins and promiscuously responds to CXCL12, CXCL12 variants, other peptides and proteins, and is relatively insensitive to mutation. To investigate the role of conformational dynamics in the distinct pharmacological behaviors of CXCR4 and ACKR3, we employed single-molecule FRET to track discrete conformational states of the receptors in real-time. The data revealed that apo-CXCR4 preferentially populates a high-FRET inactive state, while apo-ACKR3 shows little conformational preference and high transition probabilities among multiple inactive, intermediate and active conformations, consistent with its propensity for activation. Multiple active-like ACKR3 conformations are populated in response to agonists, compared to the single CXCR4 active-state. This and the markedly different conformational landscapes of the receptors suggest that activation of ACKR3 may be achieved by a broader distribution of conformational states than CXCR4. Much of the conformational heterogeneity of ACKR3 is linked to a single residue that differs between ACKR3 and CXCR4. The dynamic properties of ACKR3 may underly its inability to form productive interactions with G proteins that would drive canonical GPCR signaling.

## Introduction

CXC chemokine receptor 4 (CXCR4) is one of the most intensively studied chemokine receptors due to its central role in driving cell migration during development and immune responses, and in cancer where it promotes tumor growth and metastasis (*Chatterjee et al., 2014*; *Balkwill, 2004*; *Domanska et al., 2013*; *Kawaguchi et al., 2019*). As a class A G-protein-coupled receptor (GPCR), CXCR4 activates inhibitory Gαi protein signaling pathways to directly control cell movement in response to the chemokine CXCL12 (*Kufareva et al., 2017*). The activated receptor is also phosphorylated by GPCR kinases (GRKs), which promotes arrestin recruitment and cessation of the G protein signal. CXCR4 often works together with atypical chemokine receptor 3 (ACKR3, formerly CXCR7) which indirectly influences migration by scavenging CXCL12 to regulate available extracellular levels of the agonist and in turn the responsiveness of CXCR4 (*Griffith et al., 2014*; *Nibbs and Graham, 2013*; *Vacchini et al., 2016*). In the absence of ACKR3 scavenging, excessive CXCL12 stimulation of CXCR4 leads to downregulation, resulting in profound effects on neuronal cell migration and development (*Saaber et al., 2019*; *Lau et al., 2020*; *Wong et al., 2020*).

In contrast to CXCR4 and with some exceptions noted (*Odemis et al., 2012*; *Fumagalli et al., 2020*), ACKR3 lacks G protein activity and instead is considered to be arrestin-biased (*Rajagopal et al., 2010*). However, we and others *Saaber et al., 2019* have shown that arrestins are dispensable for chemokine scavenging while GRK phosphorylation is critical for this function (*Schafer et al., 2023*), suggesting that ACKR3 might be better described as GRK-biased. The molecular basis for its inability to couple to G proteins in most cell types remains an unanswered question. Our recently determined structures of ACKR3-ligand complexes showed the expected hallmarks of GPCR activation including 'microswitch residues' in active state configurations, displacement of transmembrane helix 6 (TM6) away from the helical bundle and an open intracellular pocket, consistent with a receptor that should be able to activate G proteins (*Yen et al., 2022*). Accordingly, we replaced the intracellular loops (ICLs) of ACKR3 with those of CXCR2, a canonical G-protein-coupled chemokine receptor; however, these changes did not lead to G protein activation (*Yen et al., 2022*), suggesting that the lack of coupling is not due to the absence of specific residue interactions. CXCL12 also adopts a distinct pose when bound to ACKR3 compared to CXCR4 and all other chemokines in chemokine-receptor complexes, but since small molecules induce similar biased effector responses, the chemokine pose cannot explain the effector-coupling bias (*Yen et al., 2022*). Having excluded other mechanisms we therefore surmised that the inability of ACKR3 to activate G proteins may be due to differences in receptor dynamics. Consistent with this hypothesis, the ICLs observed in ACKR3-agonist complexes are disordered, which may preclude productive effector coupling. The dynamic nature of ACKR3 is also suggested by its considerable constitutive activity in recruiting β-arrestins and its high level of constitutive internalization (*Yen et al., 2022*; *Hopkins et al., 2022*; *Naumann et al., 2010*).

In addition to their distinct effector interactions, ACKR3 and CXCR4 have dramatically different susceptibilities to activation by different ligands. CXCR4 is activated by a single chemokine agonist, CXCL12. Moreover, modifications of the CXCL12 N-terminal signaling domain (e.g. single point mutations as in CXCL12$_{P2G}$ or multiple mutations as in CXCL12$_{LRHQ}$) transform the chemokine into an antagonist of CXCR4 (*Hanes et al., 2015*; *Jaracz-Ros et al., 2020*). Mutational analysis, modeling, and new structures of CXCR4 suggest that ligand activation involves a precise network of interacting residues that stabilize the active receptor conformation (*Ngo et al., 2020*; *Wescott et al., 2016*; *Stephens et al., 2020*; *Liu et al., 2024*; *Saotome et al., 2025*). By contrast, ACKR3 is activated by CXCL12 as well as its variants (including CXCL12$_{P2G}$ and CXCL12$_{LRHQ}$ *Hanes et al., 2015*; *Jaracz-Ros et al., 2020*), other chemokines (CXCL11; *Burns et al., 2006*), other proteins (adrenomedullin, BAM22; *Klein et al., 2014*; *Ikeda et al., 2013*), and opioid peptides (*Meyrath et al., 2020*). In fact, most ligands for ACKR3 act as agonists, which is best explained by a non-specific 'distortion' mechanism of activation whereby any ligand that breaches the binding pocket causes helical movements that are permissive to GRK phosphorylation and arrestin recruitment. A distortion mechanism is consistent with the different binding poses observed for a small molecule agonist compared to the CXCL12 N-terminus in the receptor orthosteric pocket (*Yen et al., 2022*), and ligand bulk rather than specific interactions between agonist and ACKR3 being required for activation. We hypothesize that this distortion mechanism would also be facilitated by a receptor that is conformationally dynamic, by enabling nonspecific ligand interactions and more than a single conformation to promote activation.

To investigate the role of conformational dynamics in the distinct pharmacological behaviors of ACKR3 and CXCR4, we developed a single-molecule Förster resonance energy transfer (smFRET) approach. Many ensemble methods such as EPR, NMR and fluorescence-based methods have provided considerable insights into GPCR dynamics, conformational heterogeneity, and exchange between distinct structural states (*Liu et al., 2012*; *Wingler et al., 2019*; *Fay and Farrens, 2015*; *Yao et al., 2006*; *Elgeti and Hubbell, 2021*; *Wingler et al., 2020*; *Ray et al., 2023*). However, these methods are limited in their ability to resolve the sequence of state-to-state transitions except in rare cases where transitions can be temporally coordinated with high precision (*Schafer et al., 2016*). By contrast smFRET enables detection of sparsely populated states, reveals the sequence of state-to-state transitions and provides kinetic information through analysis of state dwell times. For example, smFRET studies of the $\beta_2$ adrenergic receptor ($\beta_2$AR) revealed a dynamic equilibrium between inactive and active conformations that was responsive to agonist and G protein binding (*Gregorio et al., 2017*). More recent smFRET studies of the glucagon receptor *Krishna Kumar et al., 2023* and the $A_{2A}$ receptor $A_{2A}$R; *Fernandes et al., 2021* have documented the existence of distinct intermediate conformations in addition to inactive and active receptor conformations.

Here, we present the first smFRET study of the chemokine receptors CXCR4 and ACKR3. Our experimental system allows real-time observation of the conformational fluctuations of individual receptor molecules in a native-like lipid environment and assessment of the differences in the conformational dynamics of the two receptors in their apo states and in response to ligands. Our results indicate that ACKR3 is more dynamic and conformationally heterogeneous than CXCR4 or other class A GPCRs previously studied, which may explain its activation prone nature and lack of G protein coupling. In contrast, CXCR4 appears less flexible, consistent with a more restricted, structurally defined activation mechanism. Together these data characterize the molecular differences between CXCR4 and ACKR3 and suggest that enhanced conformational dynamics plays an important role in the atypical function of ACKR3.

## Results

### Development of smFRET experimental systems for CXCR4 and ACKR3

To visualize conformational fluctuations of CXCR4 and ACKR3 by smFRET, cysteine residues were introduced into CXCR4 at positions $150^{4.40}$ in TM4 and $233^{6.29}$ in TM6, and at $159^{4.40}$ and $245^{6.28}$ in ACKR3 (*Figure 1—figure supplement 1A and B*) (numbers in superscript refer to the Ballesteros-Weinstein numbering scheme for GPCRs) for covalent labeling with FRET donor (D, Alexa Fluor 555, A555) and acceptor (A, Cyanine5, Cy5) fluorophores. Single receptor molecules of labeled CXCR4 or ACKR3 were reconstituted into phospholipid nanodiscs to mimic the native membrane bilayer environment. Nanodiscs have been used in a wide variety of structural and biophysical studies of integral membrane proteins, including ACKR3, and reconstitute protein-lipid interactions lost in detergent systems (*Denisov and Sligar, 2016*; *Denisov and Sligar, 2017*; *Eberle and Gustavsson, 2024*). The nanodisc-receptor complexes were then tethered to a quartz slide through biotinylation of the nanodisc membrane scaffolding protein (MSP; *Figure 1A*). To promote monomeric receptor incorporation in each nanodisc, we utilized MSP1E3D1, which forms nanodiscs with a diameter of approximately 13 nm (*Tsukamoto et al., 2010*; *Eberle and Gustavsson, 2022*).

Crystal and cryo-electron microscopy (cryo-EM) structures of homologous class A GPCRs, such as $\beta_2$AR, in inactive and active conformations, reveal that TM6 moves outwards from the TM helical bundle during activation, whereas the position of TM4 remains relatively fixed (*Rasmussen et al., 2011*). Accordingly, we anticipated that labeling at the positions indicated above would be sensitive to transitions between inactive and active receptor conformations and would give rise to different donor-acceptor distances and apparent FRET efficiencies that could be resolved by smFRET measurements (shown schematically in *Figure 1B*). These positions are similar to those used effectively for $\beta_2$AR (*Gregorio et al., 2017*), $A_{2A}$R, (*Fernandes et al., 2021*), and the μ-opioid receptor (*Zhao et al., 2024*) for monitoring their conformational dynamics by smFRET. Importantly, neither wild-type (WT) CXCR4 nor ACKR3 exhibited significant labeling, obviating the need to remove any of the native cysteine residues in either receptor. Moreover, the double cysteine receptor mutants retained CXCL12-promoted β-arrestin2 recruitment (*Figure 1—figure supplement 1C and D*). Only the $E_{max}$

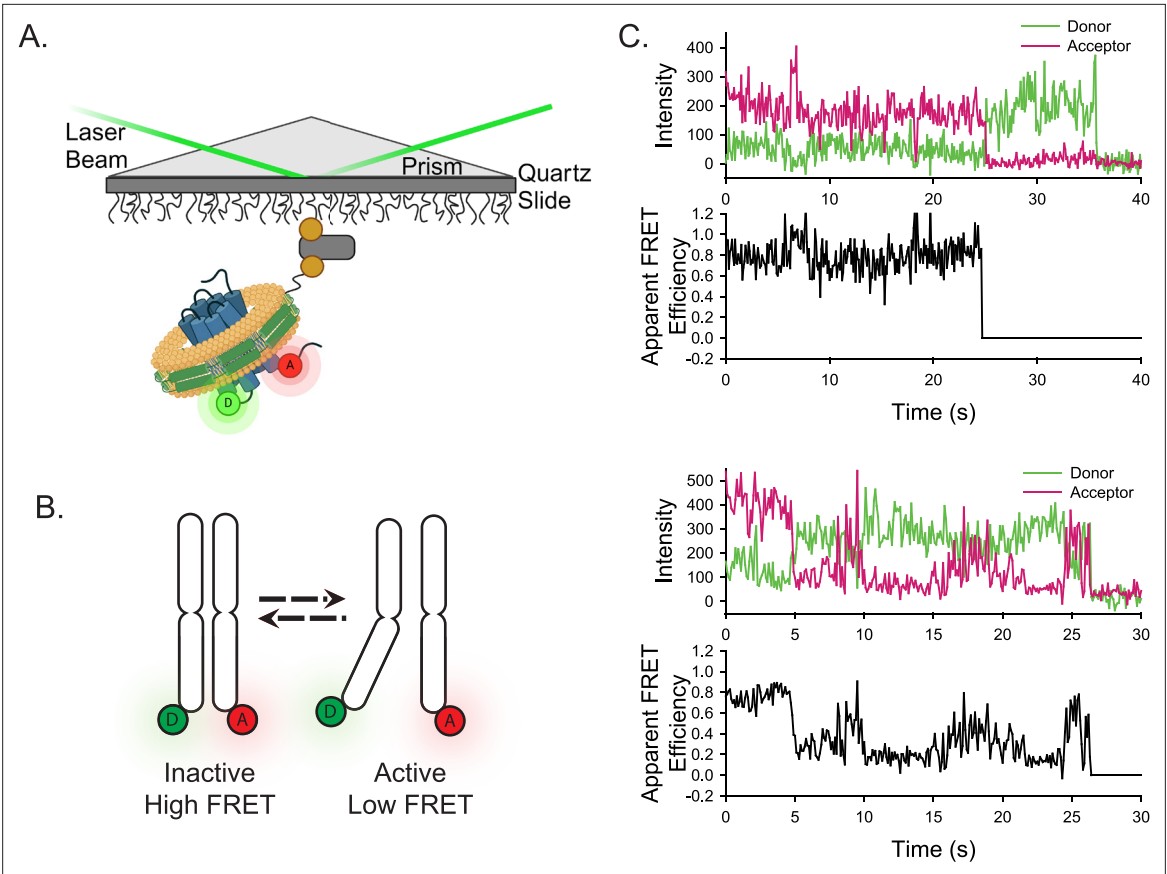

**Figure 1.** Experimental design of the smFRET system. (**A**) A single receptor molecule (blue) was labeled with donor (D) and acceptor (A) fluorophores, inserted into a phospholipid (yellow) nanodisc (green), and immobilized on a quartz slide via biotin (tan circle)-neutravidin (grey rectangle) attachment. A prism facilitates total internal reflection of the excitation laser to excite only donor fluorophores close to the surface. Created in BioRender.com. (**B**) Cartoon depicting inactive (left) and active (right) receptor conformations. (**C**) Two representative single-molecule time traces for apo-ACKR3. In both examples, the donor (green) and acceptor (red) intensities are shown in the top panel and the corresponding apparent FRET efficiency (black) is shown in the bottom panel.

The online version of this article includes the following source data and figure supplement(s) for figure 1:

**Figure supplement 1.** Cysteine mutations on TM4 and TM6 of CXCR4 and ACKR3 do not impair CXCL12 mediated β-arrestin2 recruitment observed by BRET.

**Figure supplement 1—source data 1.** Arrestin recruitment to WT and Cys-engineered CXCR4 and ACKR3 across CXCL12 concentrations.

**Figure supplement 2.** Example single-molecule traces of apo-ACKR3 and apo-CXCR4.

for arrestin recruitment to CXCL12-stimulated ACKR3 was significantly altered by the mutations, while all other pharmacological parameters were the same as for WT receptors.

## ACKR3 exhibits greater conformational dynamics than CXCR4

Receptor-nanodisc complexes were imaged on the slide surface using smFRET microscopy by exciting the A555 donor with a green (532 nm) laser and monitoring the resulting emission from both A555 and the Cy5 acceptor over time on separate segments of a CCD camera. Several hundred individual receptor-nanodisc complexes were typically observed in the field of view. Individual receptor molecules labeled with a single A555 donor and a single Cy5 acceptor were identified by single-step photobleaching transitions (abrupt loss of acceptor fluorescence signal or simultaneous loss of fluorescence signal in both channels), as shown for two representative ACKR3-nanodisc complexes in *Figure 1C*. Additionally, anti-correlated changes in donor and acceptor emission prior to photobleaching confirmed that FRET occurred between A555 and Cy5. Additional examples of single-molecule traces for ACKR3 and CXCR4 are shown in *Figure 1—figure supplement 2*. Dwell times spent in one FRET

level prior to switching to another level are generally in the seconds range (*Figure 1C*, *Figure 1—figure supplement 2*).

The apparent FRET efficiencies from many individual receptor-nanodisc complexes, recorded in the presence or absence of different ligands, were globally analyzed using Hidden Markov Models (HMMs) assuming the presence of two, three, four, or five FRET states (Materials and methods). To evaluate the appropriate level of model complexity, each resulting apparent FRET efficiency distribution was fit with a Gaussian Mixture Model (GMM) and the corresponding Bayesian Information Criterion (BIC) was calculated. The BIC is a statistical measure of the likelihood that the model describes the data, while also penalizing the addition of parameters that could lead to overfitting of noise. Theoretically, this value will be at a minimum for the model with the appropriate number of states. As an additional criterion, we carefully compared the FRET distributions following the initial HMM analysis for the different models and confirmed that peak positions were consistent across the different experimental conditions.

These analyses indicated that three FRET states were sufficient to fit the smFRET data for CXCR4 under all conditions (*Figure 2—figure supplements 1 and 2*). The resulting apparent FRET efficiency distributions are presented in *Figure 2A and B*. The predominant high-FRET state ($E_{app}$ = 0.85) observed in the apo-state (*Figure 2A*) suggests that TM4 and 6 are in close physical proximity, which is consistent with the conformation of inactive GPCRs (*Palczewski et al., 2000*; *Rasmussen et al., 2007*). Accordingly, we interpret this FRET state as the inactive conformation of CXCR4 and designate it as R. The minor low-FRET state ($E_{app}$ = 0.19) reflects outward movement of TM6 away from TM4, as expected for an active receptor conformation (designated R*) (*Rasmussen et al., 2011*; *Farrens et al., 1996*) and consistent with the limited basal activity of CXCR4. Moreover, there was a major population shift from the high-FRET state to the low-FRET state upon addition of CXCL12$_{WT}$ (*Figure 2B*), consistent with the expected stabilization of active GPCR conformations by agonists (*Kleist et al., 2022*; *Otun et al., 2024*). To gain insight into the nature of the mid-FRET state ($E_{app}$ = 0.59), we examined the connectivity between all three FRET states. Two-dimensional transition density probability (TDP) plots revealed that the three FRET states were connected in a sequential fashion (*Figure 2A and B*), indicating that the transitions occurred within the same molecules. Notably, these observations exclude the possibility that the mid-FRET state arises from different local fluorophore environments (hence FRET efficiencies) for the two possible labeling orientations of the introduced cysteines: assuming two receptor conformations, this model would produce four distinct FRET states, but only two cross peaks in the TDP plot. Another significant observation is that direct transitions between R and R* states were rarely observed (*Figure 2A and B*). Taken together, these results suggest that the mid-FRET state represents an intermediate receptor conformation (designated R') that lies on the pathway between inactive and active conformations. In the apo-state, transitions were mostly observed between states R and R' (*Figure 2A*), while in the presence of CXCL12$_{WT}$ the most frequent transitions were observed between the R' and R* states (*Figure 2B*).

In contrast, the apparent FRET efficiency histogram for ACKR3 in the apo state (*Figure 2C*) was much broader than the corresponding histogram for CXCR4 (*Figure 2A*), indicating greater conformational heterogeneity. Moreover, the smFRET data for ACKR3 could not be described by three FRET states and the inclusion of a fourth state was necessary (*Figure 2—figure supplements 3 and 4*). The FRET distributions recovered by four-state global analysis are presented in *Figure 2C and D*. The positions of the high-FRET ($E_{app}$ = 0.85) and low-FRET ($E_{app}$ = 0.11) peaks are similar to the corresponding peaks observed in CXCR4 and are likewise assigned to inactive (R) and active (R*) conformations, respectively. Consistent with these assignments, the R* state increased in population at the expense of the R state in the presence of the chemokine agonists CXCL12$_{WT}$ (*Figure 2D*) or CXCL11 (*Figure 2—figure supplement 5*). TDP plots indicated that the four FRET states in ACKR3 were connected in a sequential fashion and reside along the receptor activation pathway (*Figure 2C and D*). Similar to the arguments presented above for CXCR4, the intermediate FRET states are most likely discrete receptor conformations and not arising from mixed labeling of the two introduced cysteines. Accordingly, the mid-FRET peaks ($E_{app}$ = 0.39 and $E_{app}$ = 0.66) are assigned to two intermediate receptor conformations, designated R*' and R', respectively. Notably, the R*' state was not observed in CXCR4 (*Figure 2A and B*). The R' state in ACKR3 decreased in population in the presence of chemokine agonists (*Figure 2D*, *Figure 2—figure supplement 5B*), suggesting that this state represents an inactive intermediate receptor conformation, consistent with its position on the

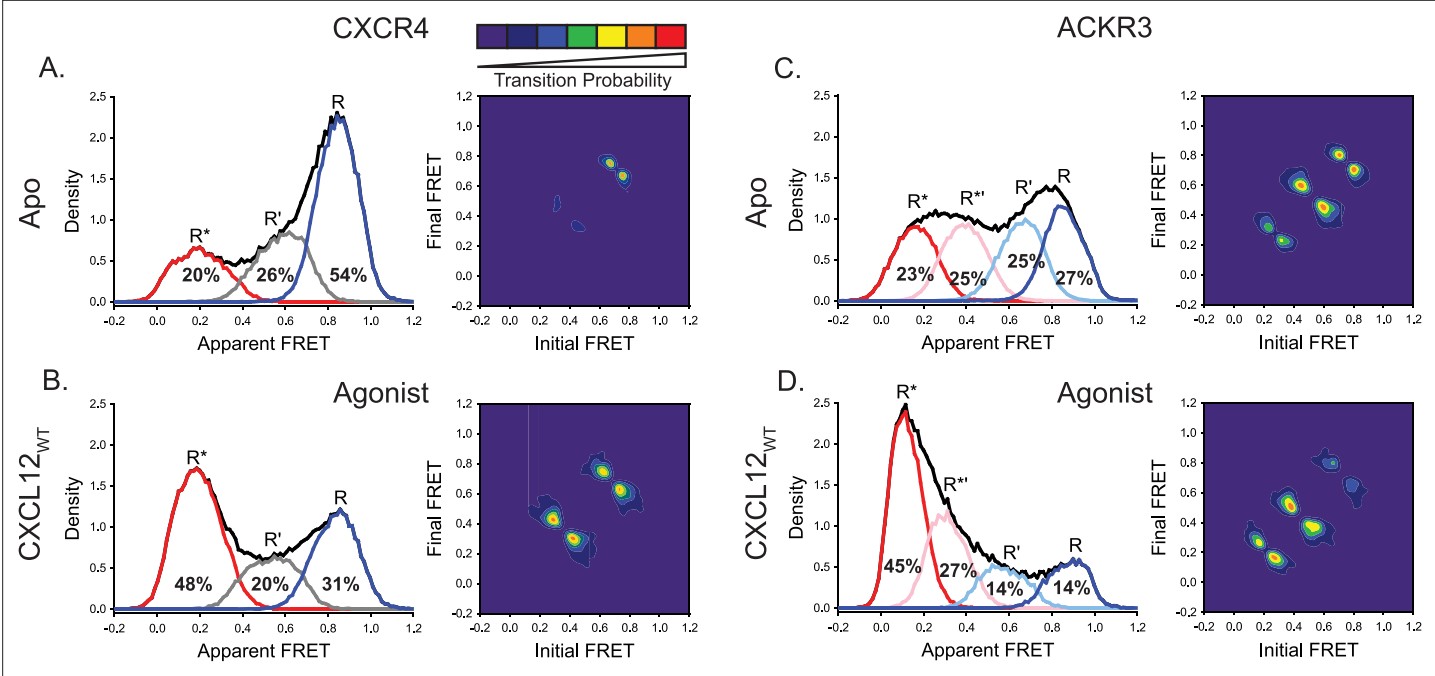

**Figure 2.** ACKR3 exhibits greater conformational flexibility compared to CXCR4. (**A**) Apparent FRET efficiency histogram of apo-CXCR4 (left, black trace) resolved into three distinct conformational states: a high-FRET state corresponding to the inactive receptor conformation (R, blue), a low-FRET active receptor conformation (R*, red) and an intermediate conformation (R', gray). The fractional populations of each state obtained from global analysis are indicated. The receptor is mostly in the inactive conformation. A transition density probability (TDP) plot (right) displays the relative probabilities of transitions from an initial FRET state (x-axis) to a final FRET state (y-axis). For apo-CXCR4, transitions between R and R' states are observed most frequently. (**B**) Addition of CXCL12$_{WT}$ to CXCR4 shifted the conformational distribution to the low-FRET R* state and resulted in more transitions between all three FRET states. (**C**) The broad apparent FRET efficiency histogram of apo-ACKR3 (left, black trace) is resolved into four distinct conformational states: inactive R (blue), active R* (red), inactive-like R' (light blue), and active-like R*' (pink). Little conformational preference is observed among these states. Moreover, all possible sequential state-to-state transitions are observed (right). (**D**) Addition of CXCL12$_{WT}$ to ACKR3 shifted the conformational distribution to the low-FRET R* state, which was also reflected in the transition probabilities. In all cases, data sets represent the analysis of at least three independent experiments.

The online version of this article includes the following source data and figure supplement(s) for figure 2:

**Source data 1.** Histograms for Apo CXCR4 FRET states.

**Source data 2.** Contour map for Apo CXCR4 TDP.

**Source data 3.** Histograms for CXCL12 CXCR4 FRET states.

**Source data 4.** Contour map for CXCL12 CXCR4 TDP.

**Source data 5.** Histograms for Apo ACKR3 FRET states.

**Source data 6.** Contour map for Apo ACKR3 TDP.

**Source data 7.** Histograms for CXCL12 ACKR3 FRET states.

**Source data 8.** Contour map for CXCL12 ACKR3 TDP.

**Figure supplement 1.** Quantitative evaluation of the appropriate number of FRET states required to model the CXCR4 smFRET distributions.

**Figure supplement 2.** Comparison of the CXCR4 intermediate states for the conditions detailed in *Figure 2—figure supplement 1* using three-state and four-state models revealed over fitting artifacts using four states.

**Figure supplement 3.** Evaluation of the number of discrete FRET states present in ACKR3 under various conditions.

**Figure supplement 4.** Comparison of intermediate state FRET histograms of ACKR3 from the three- and four-state models shown in *Figure 2—figure supplement 3*.

**Figure supplement 5.** The natural and engineered agonists CXCL11 and VUF15485, respectively, both promoted low-FRET, active-like ACKR3 conformations.

**Figure supplement 5—source data 1.** Histograms for CXCL11 ACKR3 FRET states.

**Figure supplement 5—source data 2.** Contour map for CXCL11 ACKR3 TDP.

**Figure supplement 5—source data 3.** Histograms for VUF15485 ACKR3 FRET states.

**Figure supplement 5—source data 4.** Contour map for VUF15485 ACKR3 TDP.

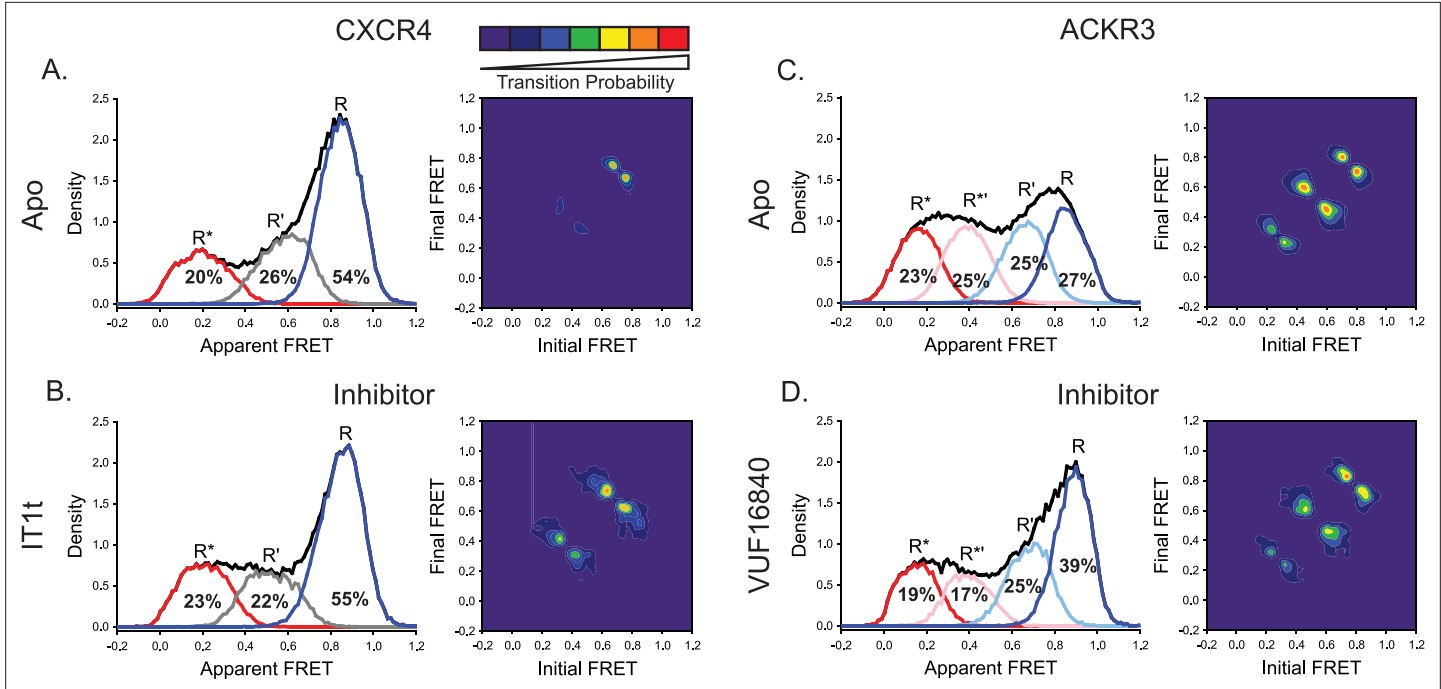

**Figure 3.** A small molecule inhibitor shifts the ACKR3 conformational population to the inactive FRET state, while CXCR4 is largely unaffected. (**A**) FRET distributions and TDP of apo-CXCR4 repeated from *Figure 2A* for comparison. (**B**) Treatment of CXCR4 with the inhibitor IT1t had little impact on the FRET distribution, but increased transition probabilities compared to the apo-receptor. (**C**) FRET distributions and TDP of apo-ACKR3 repeated from *Figure 2C* for comparison. (**D**) Treatment of ACKR3 with VUF16480, an inverse agonist, shifted the conformational distribution and TDP to the high-FRET inactive R conformation. Data sets represent the analysis of at least three independent experiments. The overall apparent FRET efficiency envelopes for the samples are represented by the black traces.

The online version of this article includes the following source data and figure supplement(s) for figure 3:

**Source data 1.** Histograms for IT1t CXCR4 FRET states.

**Source data 2.** Contour map for IT1t CXCR4 TDP.

**Source data 3.** Histograms for VUF16840 ACKR3 FRET states.

**Source data 4.** Contour map for VUF16840 ACKR3 TDP.

**Figure supplement 1.** Change in the population percentages of individual FRET states due to ligand treatment of CXCR4 and ACKR3.

**Figure supplement 1—source data 1.** Change in proportion of CXCR4 and ACKR3 FRET populations with ligand treatments.

conformational pathway (closest to R). In striking contrast to apo-CXCR4, apo-ACKR3 populated the four conformational states more or less equally, and all possible sequential conformational transitions were observed (*Figure 2C*). Thus, ACKR3 is intrinsically more conformationally heterogenous and dynamic than CXCR4. In the presence of CXCL12$_{WT}$, ACKR3 showed more frequent R' ↔ R*' and R*' ↔ R* transitions compared with the apo-receptor, accounting for the population shift towards the R* state (*Figure 2C and D*).

## Effect of small-molecule ligands on conformational states of CXCR4 and ACKR3

The small-molecule ligand IT1t is reported to act as an inverse agonist of CXCR4 (*Perpiñá-Viciano et al., 2020*; *Mona et al., 2016*; *Rosenberg et al., 2019*). However, the conformational distribution of CXCR4 showed little change to the overall apparent FRET profile, although R' ↔ R* transitions appeared in the TDP plot (*Figure 3A and B*). This suggests that the small molecule does not suppress CXCR4 basal signaling by changing the conformational equilibrium. Instead IT1t appears to increase transition probabilities which may impair G protein coupling by CXCR4.

Treatment of ACKR3 with the small-molecule agonist VUF15485 (*Zarca et al., 2024*) shifted the conformational distribution of the receptor towards the active R* state (*Figure 2—figure supplement 5C* and *Figure 3—figure supplement 1*), as expected for an agonist and supporting the

assignment of the R* FRET state. In contrast, treatment of ACKR3 with the small-molecule inverse agonist VUF16840 (*Otun et al., 2024*) shifted the conformational distribution to the inactive R conformation, with a concomitant decrease of R* (*Figure 3D*, *Figure 3—figure supplement 1*), consistent with the expected effect of an inverse agonist and with suppression of the basal activity of ACKR3 by this ligand (*Otun et al., 2024*). The intermediate R*′ population also decreased in the presence of the inverse agonist (*Figure 3D*, *Figure 3—figure supplement 1*), suggesting that this state represents an active-like receptor conformation, consistent with its placement on the conformational pathway (closest to R*). The suppression of active receptor conformations was also evident in the TDP, which revealed fewer transitions between R*′ and R* conformations relative to the apo receptor (*Figure 3C and D*).

## CXCL12 N-terminal mutants promote active receptor conformations despite their contrasting pharmacological effects on CXCR4 and ACKR3

CXCR4 is sensitive to N-terminal mutations of CXCL12 while ACKR3 is relatively insensitive. For example, the variant CXCL12$_{P2G}$, containing a proline to glycine mutation in the second position, and CXCL12$_{LRHQ}$, where the first three residues of CXCL12$_{WT}$ are replaced with the four-residue motif LRHQ starting with L0, are antagonists of CXCR4 but agonists of ACKR3 (*Hanes et al., 2015*; *Jaracz-Ros et al., 2020*). To gain further insight into the ligand-dependent responses of CXCR4 and ACKR3, we examined how these mutant chemokines influence the conformational states and dynamics of both receptors.

Surprisingly, CXCL12$_{P2G}$ promoted a shift to the active R* conformation of CXCR4 compared to the apo-receptor (R* increased by 16%, *Figure 4A and B*, *Figure 3—figure supplement 1*), although the shift was less pronounced than observed for CXCL12$_{WT}$ (28% increase in R*, *Figure 2B*, *Figure 3—figure supplement 1*). The shift is also evident in the TDPs where the state-to-state transitions involving the R* active state were more probable for the CXCL12$_{P2G}$ complex compared with apo-CXCR4 (*Figure 4A and B*). The presence of CXCL12$_{LRHQ}$ had a more subtle effect on CXCR4: the active R* conformation increased by 9% (*Figure 4A and C*, *Figure 3—figure supplement 1*). Despite the ability of CXCL12$_{P2G}$ and CXCL12$_{LRHQ}$ to stabilize the active R* conformation of CXCR4, both variants are known to act as antagonists (*Jaracz-Ros et al., 2020*). This suggests that the CXCL12 mutants inhibit CXCR4 coupling to G proteins not by suppressing the active receptor population but rather by increasing the dynamics of the receptor state-to-state transitions. Our results suggest that the helical movements considered classic signatures of the active state may not be sufficient for CXCR4 to engage productively with G proteins.

In the case of ACKR3, CXCL12$_{P2G}$ induced a modest increase in the population of the active R* conformation relative to the apo-receptor (R* increased by 5%, *Figure 4D and E*, *Figure 3—figure supplement 1*), consistent with the ability of this CXCL12 variant to act as an agonist of ACKR3 (*Jaracz-Ros et al., 2020*). Additionally, CXCL12$_{P2G}$ also promoted formation of the active-like intermediate R*′ conformation (R*′ increased by 6%, *Figure 4D and E*, *Figure 3—figure supplement 1*), suggesting that activation of ACKR3 can be achieved by populating the R*′ state, not just the R* state, which is consistent with a flexible, distortion activation mechanism. Together, CXCL12$_{P2G}$ increased active (R*) and active-like (R*′) conformations by 11%, somewhat less than observed for CXCL12$_{WT}$ (24%, *Figure 2C*, *Figure 3—figure supplement 1*). CXCL12$_{LRHQ}$ also promoted the active (R*) and active-like (R*′) conformations of ACKR3 (R*+R*′ increased by 11%, *Figure 4F*, *Figure 3—figure supplement 1*). Additionally, the probabilities of R ↔ R′ transitions were suppressed relative to apo receptor in the presence of CXCL12$_{P2G}$, while the probabilities for R′* ↔ R* transitions were enhanced in the presence of the CXCL12$_{LRHQ}$ (*Figure 4E and F*). These differences may be a consequence of the longer residence time of the LRHQ mutant on the receptor (*Gustavsson et al., 2019*). The agonism observed for both chemokine variants (*Hanes et al., 2015*; *Jaracz-Ros et al., 2020*) suggests that both the active R* and active-like R*′ conformations of ACKR3 are sufficient for GRK phosphorylation and arrestin recruitment.

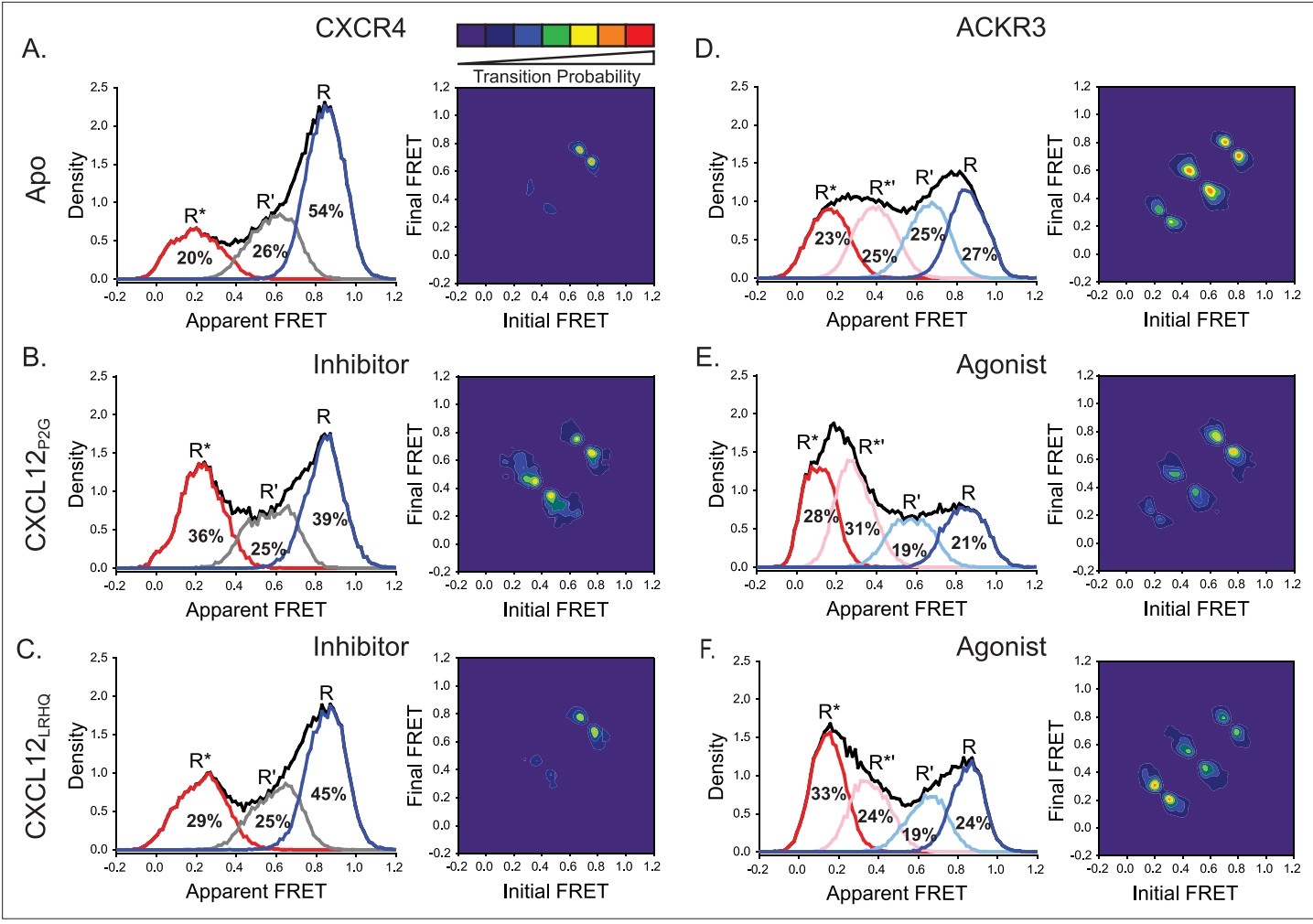

**Figure 4.** CXCL12 variants containing mutations to the N-terminus promoted active receptor conformations in both CXCR4 and ACKR3. (**A**) FRET distributions and TDP for apo-CXCR4 repeated from *Figure 2A* for reference. (**B**) Addition of CXCL12$_{P2G}$ to CXCR4 promoted a shift to the low-FRET active (R*) conformation and an increase in state-to-state transition probabilities. (**C**) CXCL12$_{LRHQ}$ led to a more subtle shift to the R* conformation of CXCR4 without affecting the transition probabilities. (**D**) FRET distributions and TDP for apo-ACKR3 repeated from *Figure 2C*. (**E**) Treatment of ACKR3 with CXCL12$_{P2G}$ displayed a shift to low-FRET, R*' and R* states, while reducing the transition probabilities for R' ↔ R*' and R*' ↔ R* transitions relative to the apo-receptor. (**F**) CXCL12$_{LRHQ}$ treatment of ACKR3 shifted the FRET distribution to the low-FRET R* active state and promoted R*' ↔ R* transitions relative to the apo-receptor. In all cases, the data sets represent the analysis of at least three independent experiments. The overall apparent FRET efficiency envelopes for the samples are represented by the black traces.

The online version of this article includes the following source data for figure 4:

**Source data 1.** Histograms for P2G CXCL12 CXCR4 FRET states.

**Source data 2.** Contour map for P2G CXCL12 CXCR4 TDP.

**Source data 3.** Histograms for LRHQ CXCL12 CXCR4 FRET states.

**Source data 4.** Contour map for LRHQ CXCL12 CXCR4 TDP.

**Source data 5.** Histograms for P2G CXCL12 ACKR3 FRET states.

**Source data 6.** Contour map for P2G CXCL12 ACKR3 TDP.

**Source data 7.** Histograms for LRHQ CXCL12 ACKR3 FRET states.

**Source data 8.** Contour map for LRHQ CXCL12 ACKR3 TDP.

## ACKR3 constitutive activity is linked to receptor conformational heterogeneity

As noted above, apo-ACKR3 displays little conformational selectivity, with similar occupancies observed for all four FRET states (*Figure 2C*). We hypothesized that this might be related to the

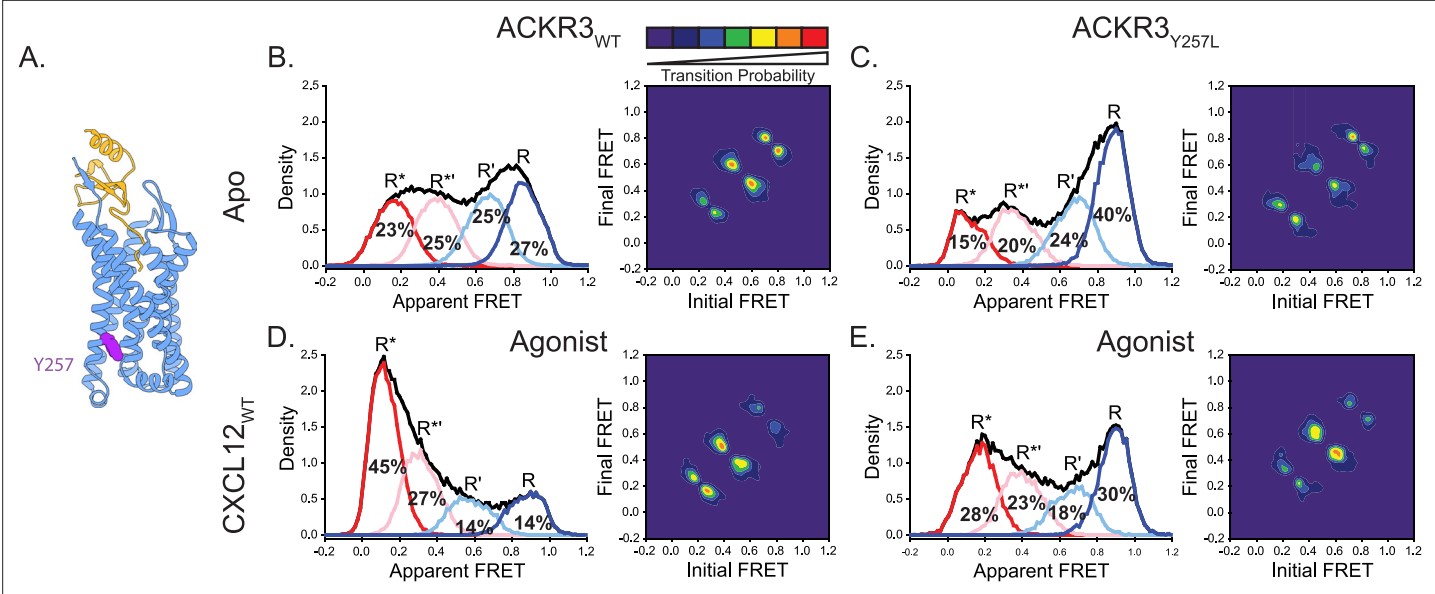

**Figure 5.** Replacement of Y257$^{6.40}$ with the corresponding residue in CXCR4 (leucine) reduces conformational heterogeneity of ACKR3. (**A**) Structure of ACKR3 bound with CXCL12$_{WT}$ (PDBID: 7SK3) highlighting the location of Y257$^{6.40}$ (purple; *Yen et al., 2022*). (**B**) Apparent FRET efficiency distributions and TDP of WT ACKR3 in the apo-state, repeated from *Figure 2C*. (**C**) The mutation Y257$^{6.40}$L shifted the conformational landscape of the apo-receptor to the high-FRET inactive R conformation at the expense of active R* and active-like R*' conformations, and also reduced the probability of state-to-state transitions. (**D**) Apparent FRET efficiency distributions and TDP of WT ACKR3 treated with CXCL12$_{WT}$, repeated from *Figure 2D*. (**E**) Treatment of Y257$^{6.40}$L ACKR3 with CXCL12$_{WT}$ promoted more low-FRET active R* and active-like R*' states. Data sets represent the analysis of at least three independent experiments. The overall apparent FRET efficiency envelopes for the samples are represented by the black traces.

The online version of this article includes the following source data and figure supplement(s) for figure 5:

**Source data 1.** Histograms for Apo Y257L ACKR3 FRET states.

**Source data 2.** Contour map for Apo Y257L ACKR3 TDP.

**Source data 3.** Histograms for CXCL12 Y257L ACKR3 FRET states.

**Source data 4.** Contour map for CXCL12 Y257L ACKR3 TDP.

**Figure supplement 1.** The mutation Y257$^{6.40}$L reduces the constitutive activity of ACKR3.

**Figure supplement 1—source data 1.** Arrestin recruitment measurements to WT and Y257L ACKR3 across CXCL12 concentrations.

constitutive activity of the receptor (*Fumagalli et al., 2020*; *Yen et al., 2022*) and tied to the presence of Tyr at position 257$^{6.40}$ (*Figure 5A*), which is a hydrophobic residue (V, I, or L) in all other chemokine receptors. In many other class A GPCRs, mutating the residue at 6.40 results in constitutive activity (*Fay and Farrens, 2015*; *Han et al., 1998*; *Han et al., 2012*; *Cui et al., 2022*) and previous analysis of rhodopsin suggests that this is a consequence of lowering the energy barrier for transitions between different receptor conformations (*Tsukamoto and Farrens, 2013*). In our previous work, we showed that mutation of Y257$^{6.40}$ to leucine, the corresponding amino acid in CXCR4, reduces constitutive arrestin recruitment to ACKR3, while preserving the ability of the receptor to be activated by CXCL12 (*Figure 5—figure supplement 1*; *Yen et al., 2022*). Adding this single-point mutation to our ACKR3 smFRET construct had a significant impact, converting the broad conformational distribution of the WT apo-receptor to a narrower distribution concentrated in the high-FRET region (*Figure 5B and C*). Overall, the FRET histogram is similar to what we observe for apo-CXCR4 (*Figure 2A*), although the active conformation is still split between R* and R*' states (the latter unique to ACKR3; *Figure 5C*). State-to-state transitions were also suppressed relative to WT ACKR3 (*Figure 5B and C*). This result indicates that Y257$^{6.40}$ is a major determinant of the broad conformational heterogeneity of ACKR3.

CXCL12$_{WT}$ promoted active and active-like conformations of Y257$^{6.40}$L ACKR3 (R*+R*' increased by 16%, *Figure 5C and E*), although the effect was somewhat reduced in comparison to WT ACKR3 (R*+R*' increased by 24%, *Figure 5B and D*). Consistent with this, the mutant receptor recruits β-arrestin in response to CXCL12$_{WT}$ with an E$_{max}$ value that is only slightly reduced relative to the WT

receptor (*Figure 5—figure supplement 1*, ~80% of WT; *Yen et al., 2022*), suggesting again that the population of the R*' intermediate conformational state may contribute to receptor activation.

## Discussion

The molecular basis for the atypical pharmacological behavior of ACKR3 is not well understood. Since structures of ACKR3 show intracellular loop disorder, and progressive structural substitutions within the loops fail to promote G protein coupling (*Yen et al., 2022*), we recently proposed that the atypical nature of ACKR3 may be related to receptor conformational dynamics (*Yen et al., 2022*; *Chen et al., 2023*). Consistent with this hypothesis, in the present study we found that the conformational dynamics of ACKR3 and the canonical GPCR CXCR4 are indeed markedly different. Our smFRET results revealed four distinct conformations of apo-ACKR3 with approximately equal populations: inactive R, active R*, R' (inactive-like) and R*' (active-like; *Figure 2C*). The state-to-state TDP plots (*Figure 2C*) further reinforced the notion that ACKR3 is a flexible receptor that readily exchanges between different conformational states, consistent with our previous structural studies where a flexible intracellular interface in the absence of interaction partners was observed (*Yen et al., 2022*). In contrast, apo-CXCR4 primarily populates the inactive R state, with only a single intermediate state (R'), and relatively few transitions between states (*Figure 2A*). These observations imply that ACKR3 has a relatively flat energy landscape, with similar energy minima for the different conformations, whereas the energy landscape of CXCR4 is more rugged (*Figure 6*). For both receptors, the energy barriers between states are sufficiently high that transitions occur relatively slowly with seconds long dwell times (*Figures 1 and 2*).

The relatively flat energy landscape of ACKR3 may account for the propensity of ACKR3 to be activated by a diverse range of ligands. In principle, ligands could promote activation by providing more energetically favorable binding interactions with the receptor in the active R* or R*' conformations relative to the inactive R or R' conformations. Alternatively, ligands could destabilize the inactive R or R' conformations of the receptor, which would also shift the receptor population to the active conformation(s). Our results do not distinguish between these two possibilities. Regardless, the similar free energies of different receptor conformations in ACKR3 (*Figure 6*) implies that relatively little energy is required to switch the receptor from one state to another. Moreover, activation can be achieved by populating either the R* or R*' state. Thus, independent of specific binding poses and receptor interactions, a variety of different ligands may be able to tip the balance between different receptor conformations.

The TDP plots demonstrate that the R' state in CXCR4 and the R' and R*' states in ACKR3 are intermediate species that lie on the pathway between inactive R and active R* receptor conformations. SmFRET studies of the glucagon receptor (*Krishna Kumar et al., 2023*) and the A2$_{AR}$ (*Fernandes et al., 2021*; both labeled in TM4 and TM6, as here) also revealed a discrete intermediate receptor conformation. In principle, the intermediates in CXCR4 and ACKR3 could represent partial movements of TM6 from the inactive to active conformation or more subtle conformational changes altering the photophysical characteristics of the probes without drastically altering the donor-acceptor distance. Either possibility leads to detectable changes in apparent FRET efficiency and reflect discrete conformational steps on the activation pathway; however, it is not possible to resolve specific structural changes from the data. Interestingly, the R*' intermediate appears to be unique to ACKR3. Moreover, this state responds to agonist and inverse agonist ligands in the same manner as the active R* state: agonists increase the population of the R*' state, while an inverse agonist decreases the population. In contrast, the single intermediate state in CXCR4 (R') is not responsive or only weakly responsive to ligands. The presence of both active (R*) and active-like (R*') conformations in ACKR3 may be responsible for its activation prone nature and ligand promiscuity.

Structural features that make ACKR3 conformationally flexible include Y257$^{6.40}$, which we previously showed contributes to the constitutive activity of ACKR3 (*Yen et al., 2022*). Similar to the constitutively active M257$^{6.40}$Y mutant of rhodopsin, Y$^{6.40}$ in ACKR3 stacks against Y$^{5.58}$ and Y$^{7.53}$ and stabilizes the active-like conformation (*Deupi et al., 2012*). Additionally, the broad conformational distribution and high probability of state-to-state transitions of ACKR3 parallel the dramatically lower energy barrier of the M257$^{6.40}$Y rhodopsin mutant relative to WT rhodopsin (*Tsukamoto and Farrens, 2013*). By contrast, mutation of Y257$^{6.40}$ to leucine, the corresponding amino acid in CXCR4, promoted the dominance of an inactive high-FRET receptor conformation similar to CXCR4 (*Figures 2A and 5C*); it

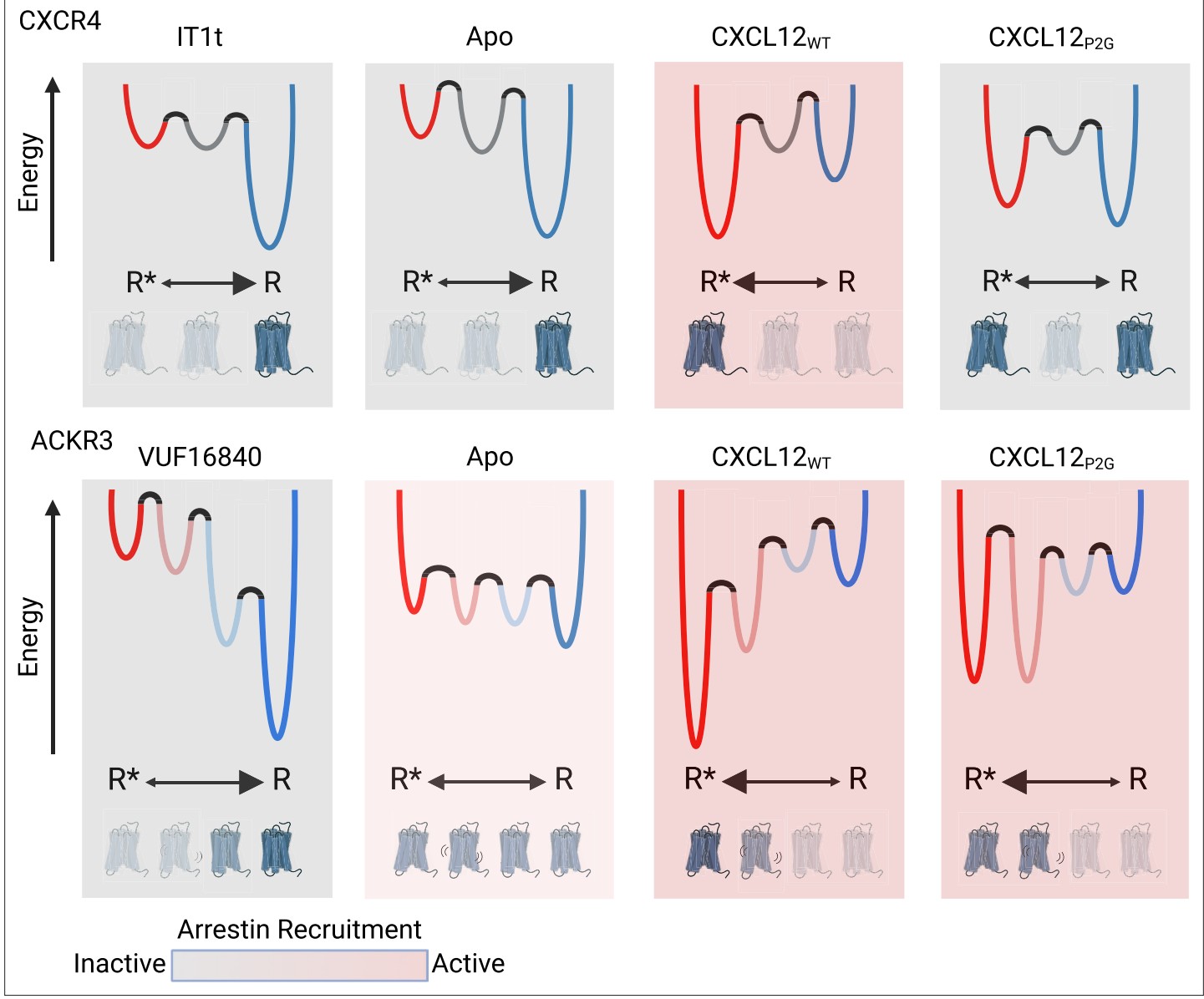

**Figure 6.** Schematic illustration of the conformational energy landscapes for CXCR4 and ACKR3, highlighting the differences in the responsiveness of the two receptors to ligands. CXCR4 populates three distinct conformations, shown here as wells on the energy landscape. Apo-CXCR4 is predominantly in the inactive R state. The receptor is converted incompletely to R* with CXCL12$_{WT}$ treatment, while the small molecule inhibitor IT1t has little impact on the conformational distribution. Though CXCL12$_{P2G}$ is an antagonist of CXCR4, the ligand promoted a detectable shift to the active R* state, suggesting TM6 movement is not sufficient for CXCR4 activation. In contrast, ACKR3 populates four distinct conformations and shows little preference among them in the apo-form. The inverse agonist, VUF16840, shifts the population to the inactive R conformation, while the agonists CXCL12$_{WT}$ and CXCL12$_{P2G}$ promote the R* and R*' populations of ACKR3. Despite stabilizing different levels of the active R* state and active-like intermediate R*' state, both CXCL12$_{WT}$ and CXCL12$_{P2G}$ are agonists of ACKR3. The flexibility of ACKR3 may contribute to the ligand-promiscuity of this atypical receptor. Figure created with BioRender.com.

The online version of this article includes the following figure supplement(s) for figure 6:

**Figure supplement 1.** Agonist-receptor interaction networks in ACKR3 and CXCR4.

also reduced the transition probabilities (*Figure 5B and C*) and the ability of ACKR3 to constitutively recruit arrestin (*Yen et al., 2022*).

In addition to the contribution of Y257$^{6.40}$ to the dynamic behavior of ACKR3, a disulfide bond between C34 in the receptor N-terminus and C287 in extracellular loop 3 (ECL3), observed in all other reported chemokine receptor-chemokine structures (*Shao et al., 2022*; *Qin et al., 2015*; *Liu et al.,*

*2020*) is conspicuously missing in cryo-EM structures of ACKR3 with chemokine and small molecule agonists (*Yen et al., 2022*). Furthermore, ACKR3 remains functional in the absence of these cysteines (*Szpakowska et al., 2018*) unlike other chemokine receptors (*Ai and Liao, 2002*; *Limatola et al., 2005*). The disulfide may constrain the relative positions of TM1 and TM6/7 and the opening of the orthosteric pocket; therefore, its absence may confer ACKR3 with greater conformational flexibility than other chemokine receptors, consistent with our observations. It may also allow ACKR3 to be activated by diverse ligands.

Our results provide insights into the linkage between receptor conformation and ligand pharmacology in CXCR4. The chemokine variants $CXCL12_{P2G}$ and $CXCL12_{LRHQ}$ are reported to act as antagonists of CXCR4 (*Hanes et al., 2015*; *Jaracz-Ros et al., 2020*), and the small molecule IT1t acts as an inverse agonist (*Mona et al., 2016*; *Rosenberg et al., 2019*; *Zarca et al., 2024*). Surprisingly, none of these ligands inhibit formation of the active R* conformation of CXCR4. In fact, the chemokine variants both stabilize and increase this state to some degree, although less effectively than $CXCL12_{WT}$. Thus, the antagonism and inverse agonism of these ligands does not appear to be linked exclusively to receptor conformation, suggesting that the ligands inhibit coupling of G proteins to CXCR4 or disrupt the ligand-receptor-G protein interaction network required for signaling (*Figure 6—figure supplement 1*; *Ngo et al., 2020*; *Stephens et al., 2020*). Interestingly, these ligands also increase the probabilities of state-to-state transitions (*Figures 3B and 4B*), suggesting that enhanced conformational exchange prevents the receptor from productively engaging G proteins. Similarly, ACKR3 is naturally dynamic and lacks G protein coupling, suggesting a common mechanism of G protein antagonism. Future smFRET studies performed in the presence of G proteins should be informative in this regard.

An unusual aspect of ACKR3 behavior is the failure to activate G proteins. Instead, ACKR3 recruits arrestins in response to phosphorylation of its C-terminal tail. Why ACKR3 does not couple to G proteins, at least in most cells, is unclear and not readily explained by differences in primary sequence, since insertion of a DRY box motif and substitution of all ICLs from a canonical GPCR failed to confer G protein activity (*Yen et al., 2022*). It is possible that the active receptor conformation clashes sterically with the G protein as suggested by docking of G proteins to structures of ACKR3 (*Yen et al., 2022*). Alternatively, the receptor dynamics and conformational transitions revealed here may prevent formation of productive contacts between ACKR3 and G protein that are required for coupling, even though G proteins appear to constitutively associate with the receptor (*Fumagalli et al., 2020*; *Yen et al., 2022*; *Levoye et al., 2009*). An important caveat is that our study does not report on the dynamics of the other TM helices and H8, some of which are known to participate in arrestin interactions (*Wingler et al., 2019*; *Fay and Farrens, 2015*). Lack of a well-organized intracellular pocket due to frequent conformational transitions may also explain why the fingerloop of arrestin is not observed to interact with the pocket, in contrast with other GPCRs (*Staus et al., 2020*; *Huang et al., 2020*), but instead inserts into membranes/micelles adjacent to the receptor (*Chen et al., 2023*). Nevertheless, arrestins are still recruited to CXCL12-stimulated ACKR3 due to GRK phosphorylation of the receptor C-terminal tail (*Schafer et al., 2023*; *Sarma et al., 2023*). Since GRKs also interact with the cytoplasmic pocket to facilitate phosphorylation, it remains to be determined how dynamics might decouple G protein activation and arrestin binding to the receptor cytoplasmic pocket (*Chen et al., 2023*) while supporting pocket-mediated GRK activity. However, given the fleeting interaction between GRKs and GPCRs (*Pulvermüller et al., 1993*), rapid state sampling by ACKR3 may not necessarily be detrimental to GRK engagement and phosphorylation. Furthermore, conformational intermediates in addition to the fully active receptor have been shown to be targets for GRK phosphorylation, such as the early photoactivated rhodopsin metarhodopsin I (*Paulsen and Bentrop, 1983*). A more constrained system may be necessary to promote productive interactions between ACKR3 and G proteins. Along these lines, a local increase of membrane pressure in certain cell environments could explain the apparent ability of ACKR3 to activate G proteins in astrocytes and glioma cells (*Odemis et al., 2012*; *Fumagalli et al., 2020*). The atypical activation of ACKR3 does not appear to be dependent on any singular receptor feature and is likely a combination of several factors.

The pharmacological behavior of ACKR3 resembles the human cytomegalovirus chemokine receptor US28, which also recognizes diverse chemokines, constitutively internalizes, and displays multiple functional conformations (*De Groof et al., 2021*). Similar to ACKR3, US28 appears to be activated by distortion of the orthosteric binding pocket rather than by specific side chain contacts

between receptor and ligand, which is supported by multiple active conformations observed for both apo-US28 and US28 with different agonists (*De Groof et al., 2021*; *Burg et al., 2015*; *Miles et al., 2018*; *Tsutsumi et al., 2022*). Whether US28 also has a relatively flat energy landscape like ACKR3 remains to be seen. The conformational dynamics and activation mechanisms revealed here for ACKR3 may also be operative in other chemokine receptors that respond to multiple ligands and have considerable constitutive activity, such as CCR1, CCR2, and CCR3 (*Shao et al., 2022*; *Gilliland et al., 2013*). Finally, the ability of ACKR3 to be activated by populating more than one conformational state may explain why antagonizing the receptor by targeting the orthosteric binding pocket has proven to be challenging; in contrast, the specific requirements for CXCR4 agonism has permitted the development of many orthosteric antagonists but few agonists (*Lefrançois et al., 2011*). Drug discovery efforts aimed at inhibiting ACKR3 may therefore require allosteric strategies.

## Materials and methods

Unless otherwise stated all chemicals and reagents were purchased from SigmaAldrich or Fisher Scientific. Methoxy e-Coelenterazine (Prolume Purple) was purchased from Nanolight Technologies (Prolume LTD).

### Cloning

Human ACKR3 (residues 2–362) preceded by an N-terminal HA signal sequence and followed by C-terminal 10His and FLAG tags was cloned into the pFasBac vector for purification from *Sf9* cells. Human CXCR4 (residues 2–352) with an N-terminal FLAG tag and C-terminal 10His was inserted into pFasBac for purification. For cell-based assays, ACKR3 (residues 2–362) or CXCR4 (residues 2–352) were inserted into pcDNA3.1 expression vector with an N-terminal FLAG tag and followed with a C-terminal *Renilla* luciferase II (ACKR3_rlucII and CXCR4_rlucII). Site-directed mutagenesis was performed by overlap extension and confirmed by Sanger sequencing. No native cysteines were substituted for either CXCR4 or ACKR3.

### Arrestin recruitment by BRET

Arrestin recruitment to ACKR3 and CXCR4 was detected using a BRET2 assay as previously described (*Schafer et al., 2023*; *Gustavsson et al., 2019*). Briefly, HEK293T cells (ATCC) were plated at 750 k/well in a six-well dish in Dulbecco's modified eagle media (DMEM) with 10% fetal bovine serum (FBS) and transfected 24 hr later with 50 ng receptor_rlucII DNA, 1 µg GFP10_β-arrestin2 (a kind gift from N. Heveker, Université de Montréal, Canada), and 1.4 µg empty pcDNA3.1 vector using TransIT-LT1 transfection system (MirusBio) and expressed for 40 hr. The cells were then washed with PBS (137 mM NaCl, 2.7 mM KCl, 10 mM $Na_2HPO_4$, 1.8 mM $KH_2PO_4$, pH 7.4) and mechanically lifted in Tyrode's buffer (25 mM HEPES, 140 mM NaCl, 1 mM $CaCl_2$, 2.7 mM KCl, 12 mM $NaHCO_3$, 5.6 mM Glucose, 0.5 mM $MgCl_2$, 0.37 mM $NaH_2PO_4$, pH 7.5). 100 k cells were plated per 96-well white BRET plate (BD Fisher) and reattached for 45 min at 37 °C. GFP expression was checked using a SpectraMax M5 plate fluorometer (Molecular Devices) with 485 nm excitation, 538 nm emission, and 530 nm cutoff. 5 µM Prolume Purple substrate was subsequently added and total luminescence detected using a TECAN Spark Luminometer (TECAN Life Sciences) at 37 °C. CXCL12 was then added to each well at the indicated final concentrations and BRET was read using default BRET2 settings (blue emission 360–440 nm, red emission 505–575 nm) and an integration time of 0.5 s. Experiments were baseline matched and normalized to the $E_{max}$ of WT receptor. The reported data is the average of three independent experiments performed in duplicate. Points were fit to a sigmoidal dose-response model using SigmaPlot 11.0 (Systat Software, Inc).

### Receptor purification, labeling, and nanodisc reconstitution

$M159^{4.40}C/Q245^{6.28}C$ ACKR3 (WT and $Y257^{6.40}L$) and $L150^{4.40}C/Q233^{6.29}C$ CXCR4 were purified from *Sf9* cells (Expression Systems) as previously described (*Yen et al., 2022*). Briefly, *Sf9* cells were infected with baculovirus (prepared using Bac-to-Bac Baculovirus Expression System, Invitrogen) containing either the mutant ACKR3 or CXCR4. Cells were harvested after 48 hr and membranes dounce homogenized in hypotonic buffer (10 mM HEPES pH 7.5, 10 mM $MgCl_2$, 20 mM KCl) followed three more times with hypotonic buffer with 1 M NaCl. The membranes were spun down at 50 k x *g* for 30 min and

resuspended between each round of douncing. After the final round, membranes were incubated with 100 µM CCX662 (Chemocentryx Inc) for ACKR3 or 100 µM IT1t for CXCR4 and solubilized in 50 mM HEPES pH 7.5, 400 mM NaCl, 0.75/0.15% dodecyl maltoside/cholesteryl hemisuccinate (DDM/CHS) with a protease inhibitor tablet (Roche) for 4 hr. Insoluble material was then removed by centrifugation at 50 k x $g$ for 30 min and Talon resin (Clontech) with 20 mM imidazole overnight binding at 4 °C. The resin was then transferred to a column and washed with WB1 (50 mM HEPES pH 7.5, 400 mM NaCl, 0.1/0.02% DDM/CHS, 10% glycerol, 20 mM imidazole) followed by WB2 (WB1 with 0.025/0.005% DDM/CHS) and finally eluted with WB2 with 250 mM imidazole. The imidazole was removed by desalting column (PD MiniTrap G-25, GE Healthcare). Final protein concentration was determined by $A_{280}$ using an extinction coefficient of 75000 $M^{-1}cm^{-1}$ (ACKR3) and 58850 $M^{-1}cm^{-1}$ (CXCR4). Samples were snap frozen in liquid nitrogen, and stored at –80 °C until use.

When ready to prepare samples for smFRET measurements, two nanomoles of receptor was thawed and incubated with fourteen nanomoles of Alexa Fluor 555 C2 maleimide (A555) and Cy5 maleimide overnight at 4 °C with rotation. The next morning, free label was removed by dilution using a 100 k Da cut-off spin concentrator (Amicon) and the sample concentrated to ~100 µl. Label incorporation was evaluated by measuring the absorbance at $A_{280}$ ($\varepsilon_{280}$=75,000 $M^{-1}cm^{-1}$ for ACKR3 and 58850 $M^{-1}cm^{-1}$ for CXCR4), $A_{555}$ ($\varepsilon_{555}$=150,000 $M^{-1}cm^{-1}$), and $A_{645}$ ($\varepsilon_{645}$=250,000 $M^{-1}cm^{-1}$) to detect the concentrations of the labeled receptor, A555, and Cy5, respectively. The contribution of the fluorophores to $A_{280}$ was removed before determining labeled receptor concentration. The entire sample was used for nanodisc reconstitution.

Labeled receptors were reconstituted into biotinylated MSP1E3D1 nanodiscs as previously described (*Yen et al., 2022*). Briefly, 1-palmitoyl-2-oleoyl-*sn*-glycero-3-phosphocholine (POPC, Avanti) and 1-palmitoyl-2-oleoyl-*sn*-glycero-3-phospho-(1'-rac-glycerol) (POPG, Avanti) were prepared in a 3:2 POPC:POPG ratio and solubilized in ND buffer (25 mM HEPES pH 7.5, 150 mM NaCl, 180 mM cholate). MSP1E3D1 was expressed and purified as previously described (*Ritchie et al., 2009*) and biotinylated with EZ-Link NHS-polyethylene glycol 4 (PEG4)-Biotin (Thermo Fisher) per manufacturer instructions. The receptors, MSP1E3D1, and lipids were combined at a molar ratio of 0.1:1:110 for ACKR3:MSP:lipids respectively. Additional ND buffer was added to keep the final cholate concentration >20 mM. After 30 min at 4 °C, 200 mg of Biobeads (Bio-Rad) were added and incubated for 3–6 hr. The sample was then loaded on a Superdex 200 10/300 GL column equilibrated with 25 mM HEPES pH 7.5, 150 mM NaCl and fractions containing nanodisc complexes were combined. 200 µl Talon resin was added with 20 mM imidazole (final concentration) and the samples were incubated for 16 hr at 4 °C. The resin was then transferred to a Micro Bio-Spin Column (Bio-Rad) and washed with 25 mM HEPES pH 7.5, 150 mM NaCl, 20 mM imidazole and eluted with 25 mM HEPES pH 7.5, 150 mM NaCl, 250 mM imidazole. Imidazole was removed by buffer-exchange using 100 k Da spin concentrators and 25 mM HEPES pH 7.5, 150 mM NaCl. Sample were concentrated to ~1 µM and stored at 4 °C until use.

## CXCL12 purification from *E. coli*

CXCL12 was expressed and purified as previously described (*Schafer et al., 2023*; *Yen et al., 2022*). Briefly, the chemokines were expressed by IPTG induction in BL21(DE3)pLysS cells. The cells were collected by centrifugation, resuspended in 50 mM tris pH 7.5, 150 mM NaCl and lysed by sonication. Inclusion bodies were then collected by centrifugation, resuspended in equilibration buffer (50 mM tris, 6 M guanidine-HCl pH 8.0), sonicated to release the chemokines and the samples centrifuged again to pellet insoluble material. The supernatant was then passed over a Ni-nitrilotri-acetic acid (NTA) column equilibrated with equilibration buffer to bind the His-tagged chemokines. The column was washed with wash buffer (50 mM MES pH 6.0, 6 M guanidine-HCl) and eluted with 50 mM acetate pH 4.0, 6 M guanidine-HCl. The chemokine-containing elutions were pooled and dithiothreitol (DTT) added to a final concentration of 4 mM. After incubating 10 min, the solution was added dropwise into refolding buffer (50 mM tris pH 7.5, 500 mM arginine-HCl, 1 mM EDTA, 1 mM oxidized glutathione) and incubated at room temperature for 4 hr before dialyzing against 20 mM tris pH 8.0, 50 mM NaCl. To remove the N-terminal purification tag, enterokinase was added and the sample incubated at 37 °C for 5 days. Uncleaved chemokine and free tags were removed by reverse Ni-NTA and eluted with wash buffer. Finally, the sample was purified on a reverse-phase C18 column equilibrated with 75% buffer A (0.1% trifluoroacetic acid (TFA)) and 25% buffer B (0.1% TFA, 90%

acetonitrile) and eluted by a linear gradient of buffer B. The pure protein was lyophilized and stored at –80 °C until use.

## smFRET microscopy

smFRET experiments were performed on a custom built prism-based TIRF microscope as previously described (*Pauszek et al., 2021*). Briefly, a flow cell was assembled on a quartz slide passivated with polyethylene glycol (PEG) and a small fraction of biotinylated PEG, after which neutravidin was introduced (*Lamichhane et al., 2010*). Labeled ACKR3 or CXCR4 in biotinylated nanodiscs were diluted into trolox buffer (25 mM HEPES pH 7.5, 150 mM NaCl, 1 mM propyl gallate, 5 mM trolox), flowed into the sample chamber, and incubated for 5 min at room temperature. The samples were then washed twice with imaging buffer (trolox buffer with 2 mM protocatechuic acid, 50 nM protocatechuate-3,4-dioxygenase). Movies were collected with 100ms integration time using a custom single-molecule data acquisition program to control the CCD camera (Andor). Single-molecule donor and acceptor emission traces were extracted from the recordings using custom IDL (Interactive Data Language) scripts. The software packages used to control the CCD camera and extract time trajectories were provided by Dr. Taekjip Ha. In all cases, five initial apo-receptor movies were recorded at different locations on the slide and then ligand was flowed into the cell by two washes with chemokine or small molecule in imaging buffer at final concentrations of 500 nM for chemokines ($CXCL12_{WT}$, $CXCL12_{P2G}$, $CXCL12_{LRHQ}$, CXCL11) and 1 µM for the small molecules (IT1t, VUF16840, VUF15485). Ten more movies were collected for each condition at different locations on the slide. The data presented are a composite of at least three individual slides and treatments.

FRET trajectories were generated and analyzed using custom software written in-house (https://github.com/rpauszek/smtirf, copy archived at *Pauszek, 2025*). Donor and acceptor traces for each molecule were corrected for donor bleed through and background signal and apparent FRET efficiencies were calculated as $E_{app} = I_A/(I_A + I_D)$, where $E_{app}$ is the apparent FRET efficiency at each time point and $I_D$ and $I_A$ are the corresponding donor and acceptor fluorophore intensities, respectively. Traces were screened manually for single donor and acceptor bleach steps and anti-correlated behavior between fluorophores to confirm the presence of single receptors within the identified particles and single donor and acceptor labeling. All traces for a particular protein/ligand combination were analyzed globally by a single Hidden Markov Model assuming two, three, four, or five states and shared variance as previously described (*Pauszek et al., 2021*). Briefly, each model was trained on all selected trajectories for a given sample simultaneously using an expectation-maximization method (*Rabiner, 1989*). Once a model was trained, the Viterbi algorithm (*Rabiner, 1989*) was used to determine the most likely hidden path for each trajectory. This labeled state path was then used to aggregate all data points belonging to a particular state in order to compile composite histograms of apparent FRET efficiency, using a Kernel Density estimation algorithm (Python package Scikit-learn, version 1.2.2) with a Gaussian kernel and a bandwidth of 0.04. The relative populations of distinct FRET states were directly obtained during compilation of the corresponding histograms. The resulting apparent FRET efficiency histograms were fit with a Gaussian Mixture Model (Scikit-learn) and the corresponding Bayesian Information Criterion (BIC) was calculated as described (*Schwarz, 1978*). Transition density probability plots revealing the connectivity among individual FRET states were calculated as described (*McKinney et al., 2006*).

## Acknowledgements

We gratefully acknowledge the initial work on this project undertaken by Chunxia Zhao and Rajan Lamichhane. Additionally, we thank Handel lab members Cheyanne Shinn, Catherina Salanga, and Nicholas Chimileski for providing the chemokines CXCL11 and $CXCL12_{P2G}$, respectively, R Leurs (Vrije Universiteit Amsterdam) for the small molecules VUF16840 and VUF15485, and N Heveker (Université de Montréal) for the GFP10_β-arrestin2 plasmid. This work was supported by National Institutes of Health grants R01 GM133157 (DPM/TMH), R01 CA254402 (TMH), R01 AI161880 (TMH), F32 GM137505 (CTS), F32 GM115017 (RFP) and T32 AI007354 (RFP). Additional support was from the Robertson Foundation/Cancer Research Institute Irvington Postdoctoral Fellowship (MG).

## Additional information

### Funding

| Funder | Grant reference number | Author |
|---|---|---|
| National Institute of General Medical Sciences | R01 GM133157 | Tracy M Handel<br>David P Millar |
| National Cancer Institute | R01 CA254402 | Tracy M Handel |
| National Institute of Allergy and Infectious Diseases | R01 AI161880 | Tracy M Handel |
| National Institute of General Medical Sciences | F32 GM137505 | Christopher T Schafer |
| National Institute of General Medical Sciences | F32 GM115017 | Raymond F Pauszek III |
| National Institute of Allergy and Infectious Diseases | T32 AI007354 | Raymond F Pauszek III |
| Robertson Foundation | Irvington Postdoctoral Fellowship | Martin Gustavsson |

The funders had no role in study design, data collection and interpretation, or the decision to submit the work for publication.

### Author contributions

Christopher T Schafer, Conceptualization, Formal analysis, Investigation, Methodology, Validation, Visualization, Writing – original draft, Writing – review and editing; Raymond F Pauszek III, Software, Formal analysis, Investigation, Validation, Visualization, Writing – review and editing; Martin Gustavsson, Visualization, Writing – review and editing; Tracy M Handel, David P Millar, Conceptualization, Supervision, Funding acquisition, Writing – original draft, Writing – review and editing

### Author ORCIDs

Christopher T Schafer ⓘ https://orcid.org/0000-0001-9907-295X
Raymond F Pauszek III, ⓘ https://orcid.org/0000-0002-3445-8429
Tracy M Handel ⓘ https://orcid.org/0000-0002-2558-6138
David P Millar ⓘ https://orcid.org/0000-0001-9207-6958

Joint public review: https://doi.org/10.7554/eLife.100098.3.sa1
Author response https://doi.org/10.7554/eLife.100098.3.sa2

## Additional files

### Supplementary files

MDAR checklist

### Data availability

All relevant data supporting this study are included in the article or source data files. FRET trajectories were generated and analyzed using custom software written in-house and available at https://github.com/rpauszek/smtirf (copy archived at *Pauszek, 2025*).

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
