## [Editor Report · eLife Assessment]

This manuscript describes the characterization of the conformational dynamics of two chemokine receptors at the single-molecule level using FRET. The authors make a **convincing** case for attributing the distinct interaction and pharmacology of the two receptors to differences in their conformational energy landscape. These **important** findings will be of interest to scientists working on activation mechanisms of GPCRs and signal transduction.

---

## [Referee Report · Joint public review]

Summary

This manuscript uses single-molecule fluorescence resonance energy transfer (smFRET) to identify differences in the molecular mechanisms of CXCR4 and ACKR3, two 7-transmembrane receptors that both respond to the chemokine CXCL12 but otherwise have very different signaling profiles. CXCR4 is highly selective for CXCL12 and activates heterotrimeric G proteins. In contrast, ACKR3 is quite promiscuous and does not couple to G proteins, but like most G protein-coupled receptors (GPCRs), it is phosphorylated by GPCR kinases and recruits arrestins. By monitoring FRET between two positions on the intracellular face of the receptor (which highlight the movement of transmembrane helix 6 [TM6], a key hallmark of GPCR activation), the authors show that CXCR4 remains mostly in an inactive-like state until CXCL12 binds and stabilizes a single active-like state. ACKR3 rapidly exchanges among four different conformations even in the absence of ligand, and agonists stabilize multiple activated states.

Strengths

The core method employed in this paper, smFRET, can reveal dynamic aspects of these receptors (the breadth of conformations explored and the rate of exchange among them) that are not evident from static structures or many other biophysical methods. smFRET has not been broadly employed in studies of GPCRs. Therefore, this manuscript makes important conceptual advances in our understanding of how related GPCRs can vary in their conformational dynamics.

Weaknesses

The probes used cannot reveal conformational changes in other positions besides transmembrane helix 6 (TM6). GPCRs are known to exhibit loose allosteric coupling, so the conformational distribution observed at TM6 may not fully reflect the global conformational distribution of receptors. This could mask important differences that determine the ability of intracellular transducers to couple to specific receptor conformations.

While it is clear that CXCR4 and ACKR3 have very different conformational dynamics, the data do not definitely show that this is the main or only mechanism that contributes to their functional differences.

The extent to which conformational heterogeneity is a characteristic feature of ACKRs that contributes to their promiscuity and arrestin bias is unclear. The key residue the authors find promotes ACKR3 conformational heterogeneity is not conserved in most other ACKRs, but alternative mechanisms could generate similar heterogeneity.

An inherent limitation of the approach is that mutagenesis, purification, and labeling of the receptors could affect their conformational distributions. The cysteine mutations in ACKR3 required to site-specifically install fluorophores substantially increase its ligand-induced activity (Fig. S1D). There are no data to confirm that the two receptors retain the same functional profiles observed in cell-based systems following in vitro manipulations (purification, labeling, nanodisc reconstitution).

---

## [Author Response]

The following is the authors’ response to the original reviews.

**Reviewer #1 (Public Review):**
Summary:This paper uses single-molecule FRET to investigate the molecular basis for the distinct activation mechanisms between 2 GPCR responding to the chemokine CXCL12 : CXCR4, that couples to G-proteins, and ACKR3, which is G-protein independent and displays a higher basal activity.Strengths:It nicely combines the state-of-the-art techniques used in the studies of the structural dynamics of GPCR. The receptors are produced from eukaryotic cells, mutated, and labeled with single molecule compatible fluorescent dyes. They are reconstituted in nanodiscs, which maintain an environment as close as possible to the cell membrane, and immobilized through the nanodisc MSP protein, to avoid perturbing the receptor's structural dynamics by the use of an antibody for example.The smFRET data are analysed using the HHMI technique, and the number of states to be taken into account is evaluated using a Bayesian Information Criterion, which constitutes the state-of-the-art for this task.The data show convincingly that the activation of the CXCR4 and ACKR3 by an agonist leads to a shift from an ensemble of high FRET states to an ensemble of lower FRET states, consistent with an increase in distance between the TM4 and TM6. The two receptors also appear to explore a different conformational space. A wider distribution of states is observed for ACKR3 as compared to CXCR4, and it shifts in the presence of agonists toward the active states, which correlates well with ACKR3's tendency to be constitutively active. This interpretation is confirmed by the use of the mutation of Y254 to leucine (the corresponding residue in CXCR4), which leads to a conformational distribution that resembles the one observed with CXCR4. It is correlated with a decrease in constitutive activity of ACKR3.Weaknesses:Although the data overall support the claims of the authors, there are however some details in the data analysis and interpretation that should be modified, clarified, or discussed in my opinionConcerning the amplitude of the changes in FRET efficiency: the authors do not provide any structural information on the amplitude of the FRET changes that are expected. To me, it looks like a FRET change from ~0.9 to ~0.1 is very important, for a distance change that is expected to be only a few angstroms concerning the movement of the TM6. Can the authors give an explanation for that? How does this FRET change relate to those observed with other GPCRs modified at the same or equivalent positions on TM4 and TM6?

The large FRET change in our system was initially unexpected. However, the reviewer is mistaken that the expected distance change is only a few angstroms. Crystal structures of the homologous beta2 adrenergic receptor (β_2_AR) in inactive and active conformations reveal that the cytoplasmic end of TM6 moves outwards by 16 angstroms during activation (Rasmussen et al., 2011). Consistent with this, smFRET studies of β_2_AR labeled in TM4 and TM6 (as here) showed that the donor-acceptor (D-A) distance was 14 angstroms longer in the active conformation (Gregorio et al., 2017). Surprisingly, the apparent distance change in our system (calculated for our FRET probes, A555/Cy5, using FPbase.com) is almost 30 angstroms. A possible explanation is that the fluorophore attached to TM6 interacts with lipids within the nanodisc when TM6 moves outwards, which could stretch the fluorophore linker and thereby increase the D-A distance (lipids were absent in the β_2_AR study). Such an interaction could also constrain the fluorophore in an unfavorable orientation for energy transfer, also leading to lower than expected FRET efficiencies and inflated distance calculations. Regardless, it is important to emphasize that none of the interpretations or conclusions of our study are based on computed D-A distances. Rather, we resolved different receptor conformations and quantified their relative populations based on the measured FRET efficiency distributions.

Finally, we note that a recent smFRET study of the glucagon receptor (labeled in TM4 and TM6, as here) also revealed a large difference in apparent FRET efficiencies between inactive (*E*_app_ = 0.83) and active (*E*_app_ = 0.32) conformations (Krishna Kumar et al., 2023). Thus, the large change in FRET efficiency observed in our study is not unprecedented.

Concerning the intermediate states: the authors observe several intermediate states.(1) First I am surprised, looking at the time traces, by the dwell times of the transitions between the states, which often last several seconds. Is such a long transition time compatible with what is known about the kinetic activation of these receptors?

We too were surprised by the apparent kinetics of the receptors in our system. However, it was previously noted that purified systems, including nanodiscs, lead to slower activation times for GPCRs compared to cellular membrane systems (Lohse et al, Curr. Opin. Cell Biology, 27, 8792, 2014). Indeed, slow transitions among different FRET states (dwell times in the seconds range) were also observed in recent smFRET studies of the mu opioid receptor (Zhao et al., 2024) and the glucagon receptor (Krishna Kumar et al., 2023). These studies are consistent with the observed time scale of the FRET transitions reported here.

(2) Second is it possible that these “intermediate” states correspond to differences in FRET efficiencies, that arise from different photophysical states of the dyes? Alexa555 and Cy5 are Cyanines, that are known to be very sensitive to their local environment. This could lead to different quantum yields and therefore different FRET efficiencies for a similar distance. In addition, the authors use statistical labeling of two cysteines, and have therefore in their experiment a mixture of receptors where the donor and acceptor are switched, and can therefore experience different environments. The authors do not speculate structurally on what these intermediate states could be, which is appreciated, but I think they should nevertheless discuss the potential issue of fluorophore photophysics effects.

The reviewer is correct that the intermediate FRET states could, in principle, arise from a conformational change of the receptor that alters the local environment of the donor and/or acceptor fluorophores, rather than a change in donor-acceptor distance. This caveat is now included in the discussion on Pg. 10:

“In principle, the intermediates in CXCR4 and ACKR3 could represent partial movements of TM6 from the inactive to active conformation or more subtle conformational changes altering the photophysical characteristics of the probes without drastically altering the donor-acceptor distance. Either possibility leads to detectable changes in apparent FRET efficiency and reflect discrete conformational steps on the activation pathway; however, it is not possible to resolve specific structural changes from the data.”

Regarding the second possibility, it is true that our labeling methodology leads to a statistical mixture of labeled species (D on TM6 and A on TM4, D on TM4 and A on TM6). If the photophysical properties of the fluorophores were markedly different for the two labeling orientations, this would produce two different FRET efficiencies for a given receptor conformation. Assuming two receptor conformations, this scenario would produce four distinct FRET states: *E*_1_ (inactive receptor, labeling configuration 1), *E*_2_ (active receptor, labeling configuration 1), *E*_3_ (inactive receptor, labeling configuration 2) and *E*_4_ (active receptor, labeling configuration 2), with two cross peaks in the TDP plots, corresponding to *E*_1_ ↔ *E*_2_ and *E*_3_ ↔ *E*_4_ transitions. Notably, *E*_2_ ↔ *E*_3_ cross peaks would not be present, since states *E*_2_ and *E*_3_ exist on separate molecules. Instead, we see all states inter-connected sequentially, R ↔ R’ ↔ R* in CXCR4 and R ↔ R’ ↔ R*’ ↔ R* in ACKR3 (Fig. 2), suggesting that the resolved FRET states represent interconnected conformational states.

We added the following text to the Results section on Pg. 6:

“Two-dimensional transition density probability (TDP) plots revealed that the three FRET states were connected in a sequential fashion (Figs. 2A & B), indicating that the transitions occurred within the same molecules. Notably, these observations exclude the possibility that the midFRET state arises from different local fluorophore environments (hence FRET efficiencies) for the two possible labeling orientations of the introduced cysteines: assuming two receptor conformations, this model would produce four distinct FRET states, but only two cross peaks in the TDP plot.”

(3) It would also have been nice to discuss whether these types of intermediate states have been observed in other studies by smFRET on GPCR labeled at similar positions.

Intermediate states have also been reported in previous smFRET studies of other GPCRs. For example, in the glucagon receptor (also labeled in TM4 and TM6), a third FRET state (*E*_app_ = 0.63) was resolved between the inactive (*E*_app_ = 0.85) and active (*E*_app_ = 0.32) states (Krishna Kumar et al., 2023). Discrete intermediate receptor conformations were also observed in the A_2A_R labeled in TM4 and TM6 (Fernandes et al., 2021). These examples are now cited in the Discussion.

On line 239: the authors talk about the R↔R' transitions that are more probable. In fact it is more striking that the R'↔R* transition appears in the plot. This transition is a signature of the behavior observed in the presence of an agonist, although IT1t is supposed to be an inverse agonist. This observation is consistent with the unexpected (for an inverse agonist) shift in the FRET histogram distribution. In fact, it appears that all CXCR4 antagonists or inverse agonists have a similar (although smaller) effect than the agonist. Is this related to the fact that these (antagonist or inverse agonist) ligands lead to a conformation that is similar to the agonists, but cannot interact with the G-protein ?? Maybe a very interesting experiment would be here to repeat these measurements in the presence of purified G-protein. G-protein has been shown to lead to a shift of the conformational space explored by GPCR toward the active state (using smFRET on class A and class C GPCR). It would be interesting to explore its role on CXCR4 in the presence of these various ligands. Although I am aware that this experiment might go beyond the scope of this study, I think this point should be discussed nevertheless.

We thank the reviewer for this observation and the possible explanation offered. In response, we have added the following text to the Results section on Pg. 7:

“The small-molecule ligand IT1t is reported to act as an inverse agonist of CXCR4 (54-56). However, the conformational distribution of CXCR4 showed little change to the overall apparent

FRET profile, although R’ ↔ R* transitions appeared in the TDP plot (Figs. 3A & B, Fig. S8). This suggests that the small molecule does not suppress CXCR4 basal signaling by changing the conformational equilibrium. Instead IT1t appears to increase transition probabilities which may impair G protein coupling by CXCR4.”

We have also added the following text to the Results on Pg. 8:

“Despite the ability of CXCL12_P2G_ and CXCL12_LRHQ_ to stabilize the active R* conformation of CXCR4, both variants are known to act as antagonists (20). This suggests that the CXCL12 mutants inhibit CXCR4 coupling to G proteins not by suppressing the active receptor population but rather by increasing the dynamics of the receptor state transitions. Our results suggest that the helical movements considered classic signatures of the active state may not be sufficient for CXCR4 to engage productively with G proteins.”

In addition, we have added the following text to the Discussion on Pg. 11:

“The chemokine variants CXCL12_P2G_ and CXCL12_LRHQ_ are reported to act as antagonists of CXCR4 (19, 20), and the small molecule IT1t acts as an inverse agonist (54-56). Surprisingly, none of these ligands inhibit formation of the active R* conformation of CXCR4. In fact, the chemokine variants both stabilize and increase this state to some degree, although less effectively than CXCL12_WT_. Thus, the antagonism and inverse agonism of these ligands does not appear to be linked exclusively to receptor conformation, suggesting that the ligands inhibit coupling of G proteins to CXCR4 or disrupt the ligand-receptor-G protein interaction network required for signaling (Fig. S10) (21, 23). Interestingly, these ligands also increase the probabilities of state-to-state transitions (Figs. 3B & 4B), suggesting that enhanced conformational exchange prevents the receptor from productively engaging G proteins. Similarly, ACKR3 is naturally dynamic and lacks G protein coupling, suggesting a common mechanism of G protein antagonism.”

Finally, we also agree that experiments with G proteins could be informative. In fact, we initiated such experiments during the course of this study. However, it soon became apparent that significant optimization would be required to identify fluorophore labeling positions that report receptor conformation without inhibiting G protein coupling. Accordingly, we decided that G protein experiments would be the subject of future studies.

However, we added the following text to the Discussion on Pg. 12:

“Future smFRET studies performed in the presence of G proteins should be informative in this regard”.

The authors also mentioned in Figure 6 that the energetic landscape of the receptors is relatively flat ... I do not really agree with this statement. For me, a flat conformational landscape would be one where the receptors are able to switch very rapidly between the states (typically in the submillisecond timescale, which is the timescale of protein domain dynamics). Here, the authors observed that the transition between states is in the second timescale, which for me implies that the transition barrier between the states is relatively high to preclude the fast transitions.

We thank the reviewer for the comment. We have modified the description of the energy landscapes of ACKR3 and CXCR4 in the discussion on Pg. 10 as follows:

“These observations imply that ACKR3 has a relatively flat energy landscape, with similar energy minima for the different conformations, whereas the energy landscape of CXCR4 is more rugged (Fig. 6). For both receptors, the energy barriers between states are sufficiently high that transitions occur relatively slowly with seconds long dwell times (Figs. 1C and S2).”

**Reviewer #2 (Public Review):**
Summary:his manuscript uses single-molecule fluorescence resonance energy transfer (smFRET) to identify differences in the molecular mechanisms of CXCR4 and ACKR3, two 7transmembrane receptors that both respond to the chemokine CXCL12 but otherwise have very different signaling profiles. CXCR4 is highly selective for CXCL12 and activates heterotrimeric G proteins. In contrast, ACKR3 is quite promiscuous and does not couple to G proteins, but like most G protein-coupled receptors (GPCRs), it is phosphorylated by GPCR kinases and recruits arrestins. By monitoring FRET between two positions on the intracellular face of the receptor (which highlights the movement of transmembrane helix 6 [TM6], a key hallmark of GPCR activation), the authors show that CXCR4 remains mostly in an inactive-like state until CXCL12 binds and stabilizes a single active-like state. ACKR3 rapidly exchanges among four different conformations even in the absence of ligands, and agonists stabilize multiple activated states.Strengths:The core method employed in this paper, smFRET, can reveal dynamic aspects of these receptors (the breadth of conformations explored and the rate of exchange among them) that are not evident from static structures or many other biophysical methods. smFRET has not been broadly employed in studies of GPCRs. Therefore, this manuscript makes important conceptual advances in our understanding of how related GPCRs can vary in their conformational dynamics.Weaknesses:(1) The cysteine mutations in ACKR3 required to site-specifically install fluorophores substantially increase its basal and ligand-induced activity. If, as the authors posit, basal activity correlates with conformational heterogeneity, the smFRET data could greatly overestimate the conformational heterogeneity of ACKR3.

The change in basal ACKR3 activity with the Cys introductions are modest in comparison and insignificantly different as determined by extra-sum-of-squares F test (P=0.14).

(2) The probes used cannot reveal conformational changes in other positions besides TM6. GPCRs are known to exhibit loose allosteric coupling, so the conformational distribution observed at TM6 may not fully reflect the global conformational distribution of receptors. This could mask important differences that determine the ability of intracellular transducers to couple to specific receptor conformations.

We agree that the overall conformational landscape of the receptors has not been investigated and we have added this caveat to the discussion on Pg. 12.

“An important caveat is that our study does not report on the dynamics of the other TM helices and H8, some of which are known to participate in arrestin interactions.”

(3) While it is clear that CXCR4 and ACKR3 have very different conformational dynamics, the data do not definitively show that this is the main or only mechanism that contributes to their functional differences. There is little discussion of alternative potential mechanisms.

The main functional difference between CXCR4 and ACRK3 is their effector coupling: CXCR4 couples to G proteins, whereas ACKR3 only couples to arrestins (following phosphorylation of the C-terminal tail by GRKs). As currently noted in the discussion, ACKR3 has many features that may contribute to its lack of G protein coupling, including lack of a well-ordered intracellular pocket due to conformational dynamics, lack of an N-term-ECL3 disulfide, different chemokine binding mode, and the presence of Y257. Steric interference due to different ICL loop structures may also interfere with G protein activation. No one thing has proven to confer ACKR3 with G protein activity including swapping all of the ICLs to those of canonical chemokine receptor, suggesting it is a combination of these different factors. The following has been added to the discussion on Pg. 13 to clearly note that any one feature is unlikely to drive the atypical behavior of ACKR3:

“The atypical activation of ACKR3 does not appear to be dependent on any singular receptor feature and is likely a combination of several factors.”

(4) The extent to which conformational heterogeneity is a characteristic feature of ACKRs that contributes to their promiscuity and arrestin bias is unclear. The key residue the authors find promotes ACKR3 conformational heterogeneity is not conserved in most other ACKRs, but alternative mechanisms could generate similar heterogeneity.

Despite the commonalities in the roles of the ACKRs, they all appear to have evolved independently. Thus, we do not believe that all features observed and described for one ACKR will explain the behavior of another. We have carefully avoided expanding our observations to other ACKRs to avoid suggesting common mechanisms.

(5) There are no data to confirm that the two receptors retain the same functional profiles observed in cell-based systems following in vitro manipulations (purification, labeling, nanodisc reconstitution).

We agree this is an important point. All labeled receptors responded to agonist stimulation as expected. As only properly folded receptors are able to make the extensive interactions with ligands necessary for conformational changes (for instance, CXCL12 interacts with all TMs and ECLs), this suggests that the proteins are folded correctly and functional following all manipulations.

**Reviewer #3 (Public Review):**
Summary:This is a well-designed and rigorous comparative study of the conformational dynamics of two chemokine receptors, the canonical CXCR4 and the atypical ACKR3, using single-molecule fluorescence spectroscopy. These receptors play a role in cell migration and may be relevant for developing drugs targeting tumor growth in cancers. The authors use single-molecule FRET to obtain distributions of a specific intermolecular distance that changes upon activation of the receptor and track differences between the two receptors in the apo state, and in response to ligands and mutations. The picture emerging is that more dynamic conformations promote more basal activity and more promiscuous coupling of the receptor to effectors.Strengths:The study is well designed to test the main hypothesis, the sample preparation and the experiments conducted are sound and the data analysis is rigorous. The technique, smFRET, allows for the detection of several substates, even those that are rarely sampled, and it can provide a "connectivity map" by looking at the transition probabilities between states. The receptors are reconstituted in nanodiscs to create a native-like environment. The examples of raw donor/acceptor intensity traces and FRET traces look convincing and the data analysis is reliable to extract the sub-states of the ensemble. The role of specific residues in creating a more flat conformational landscape in ACKR3 (e.g., Y257 and the C34-C287 bridge) is well documented in the paper.Weaknesses:The kinetics side of the analysis is mentioned, but not described and discussed. I am not sure why since the data contains that information. For instance, it is not clear if greater conformational flexibility is accompanied by faster transitions between states or not.

The reviewer is correct that kinetic information is available, in principle, from smFRET experiments. However, a detailed kinetic analysis will require a much larger data set than we currently possess, to adequately sample all possible transitions and the dwell times of each FRET state. We intend to perform such an analysis in the future as more data becomes available. The purpose of this initial study was to explore the conformational landscapes of CXCR4 and ACKR3 and to reveal differences between them. To this end, we have documented major differences in conformational preferences and response to ligands of the two receptors that are likely relevant to their different biological behavior. Future kinetic information will add further detail, but is not expected to alter the conclusions drawn here.

The method to choose the number of states seems reasonable, but the "similarity" of states argument (Figures S4 and S6) is not that clear.

We thank the reviewer for noting a need for further clarification. We qualitatively compared the positions of the various FRET peaks across treatments to gain insight into the consistency of the conformations and avoid splitting real states by overfitting the data. For instance, fitting the ACKR3 treatments with three states leads to three distinct FRET populations for the R’ intermediate. Adding a fourth state results in two intermediates that are fairly well overlapping. In contrast, the two-intermediate model for CXCR4 appears to split the R* state of the CXCL12 treated sample and causes a general shift in both intermediate states to lower FRET values when CXCL12 is present. As we assume that the conformations are consistent throughout the treatments, we conclude that this represents an overfitting artifact and not a novel CXCR4 R*’ state. Additional sentences have been added to the supplemental figure legend to better describe the comparative analysis.

“(Top) With the 3-state model, the R’ states for apo-CXCR4 and for CXCL12- and IT1t-bound receptor overlapped well with similar apparent FRET values across all of the tested conditions. In the case of the four-state model, the R*’ (Middle) and R’ (Bottom) states were substantially different across the ligand treatments. In particular, the R*’ state with CXCL12 treatment appears to arise from a splitting of the R* conformation, indicating that the model was overfitting the data.”

Also, the "dynamics" explanation offered for ACKR3's failure to couple and activate G proteins is not very convincing. In other studies, it was shown that activation of GPCRs by agonists leads to an increase in local dynamics around the TM6 labelling site, but that did not prevent G protein coupling and activation.

We agree with the reviewer that any single explanation for ACKR3 bias, including the dynamics argument presented here, is insufficient to fully characterize the ACKR3 responses. As noted by the reviewer, the TM6 movement and dynamics is generally correlated with G protein coupling, whereas other dynamics studies (Wingler et al. 2019) have noted that arrestinbiased ligands do not lead to the same degree of TM6 movement. We have added the following statement to the discussion on Pg. 13:

“The atypical activation of ACKR3 does not appear to be dependent on any singular receptor feature and is likely a combination of several factors.”

**Recommendations for the authors:**

**Reviewer #1 (Recommendations For The Authors):**
I would like to raise a technical point about the calculation and reporting of the FRET efficiency. The authors report the FRET efficiency as E=IA/(IA+ID). There is now a strong recommendation from the FRET community (https://doi.org/10.1038/s41592-018-0085-0) to use the term “FRET efficiency” only when a proper correction procedure of all correction factors has been applied, which is not the case here (gamma factor has not been calculated). The authors should therefore use the term “Apparent FRET Efficiency” and *E*_app_ in all the manuscripts.Also, it would be nice to indicate directly on the figures whether a ligand that is used is an agonist, antagonist, inverse agonist, etc...

We thank the reviewer for suggesting this clarification in terminology. We now refer to apparent FRET efficiency (or *E*_app_) throughout the manuscript and in the figures. In addition, we have added ligand descriptions to the relevant figures.

**Reviewer #2 (Recommendations For The Authors):**
(1) M159(4.40)C/Q245(6.28)C ACKR3 appears to have higher constitutive activity than ACKR3 Wt (Fig. S1). While the vehicle point itself is likely not significant due to the error in the Wt, the overall trend is clear and arguably even stronger than the effect of Y257(6.40)L (Fig. S9). While this is an inherent limitation of the method used, it should be clearly acknowledged; the comment in lines 162-164 seems to skirt the issue by only saying that arrestin recruitment is retained. It would be helpful and more rigorous to report the curve fit parameters (basal, E_max_, EC50) for the arrestin recruitment experiments and the associated errors/significance (see https://www.graphpad.com/guides/prism/latest/statistics/stat_qa_multiple_comparisons_after_.htm for a discussion).

The Emin, E_max_, and EC50 for M159^4^.^40^C/Q245^6^.^28^C ACKR3 were compared against the values for WT ACKR3 from Fig. S1 and only the E_max_ was determined to be significantly different by the extra sum of squares F test. A note has been added to the text to reflect these results on Pg. 5.

“Only the E_max_ for arrestin recruitment to CXCL12-stimulated ACKR3 was significantly altered by the mutations, while all other pharmacological parameters were the same as for WT receptors.”

(2) The methods do not specify the reactive group of the dyes used for labeling (i.e., AlexaFluor 555-maleimide and Cy5-maleimide?).

We regret the omission and have added the necessary details to the materials and methods.

(3) Were any of the native Cys residues removed from ACKR3 and CXCR4 in the constructs used for smFRET? ACKR3 appears to have two additional Cys residues in the N-terminus besides the one involved in the second disulfide bridge, and these would presumably be solvent-exposed. If so, please specify in the Methods and clarify whether the constructs tested in functional assays included these. (Also, please specify if the human receptors were used.)

No additional cysteine residues were mutated in either receptor. All exposed cysteines are predicted to form disulfides. The residues in the N-terminus that the reviewer alludes to, C21 and C26, form a disulfide (Gustavsson et al. Nature Communications 8, 14135, 2017) and are thus protected from our probes. Consistent with these expectations, neither WT CXCR4 nor ACKR3 exhibited significant fluorophore labeling (now mentioned in the text on Pg. 5). The species of origin has been added to the material and methods.

(4) There are a few instances where the data seem to slightly diverge from the proposed models that may be helpful to comment on explicitly in the text:- Figure 4E (ACKR3/CXCL12(P2G)): As noted in the legend, despite stabilizing R*/R*', CXCL12(P2G) reduces transitions between these states compared to Apo. This is more similar to the effects of VUF16840 (Figure 3D) than the other ACKR3 agonists. The authors note the difference between CXCL12(LHRQ) and CXCL12(P2G) (but not vs Apo) in this regard. There might be some other information here regarding the relative importance of the conformational equilibrium vs transition rates for receptor activity.

Although the TDPs for CXCL12_P2G_ and VUF16840 are similar, as noted by the reviewer, the overall FRET envelopes are drastically different.

The differences in transition probabilities for R ↔ R’ and R*’ « R* transitions observed in the presence of CXCL12_P2G_ or CXCL12_LRHQ_ relative to the apo receptor are now explicitly noted in the Results.

- The conformational distributions of ACKR3 apo and ACKR3 Y257L CXCL12 are very similar (Figure 5A,D). However, there is a substantial difference in the basal activity of WT vs CXCL12stimulated Y257L (Figure S9).

The mutation Y257L appears to promote the highest and lowest FRET states at the expense of the intermediates. Although the distribution appears similar between Apo-WT and CXCL12Y257L, the depopulation of the R’ state may lead to the observed activation in cells.

(5) There are inconsistent statements regarding the compatibility of G protein binding to the "active-like" ACKR3 conformation observed in the authors' previous structures (Yen et al, Sci Adv 2022). In the introduction, the authors seem to be making the case that steric clashes cannot account for its lack of coupling; in the discussion, they seem to consider it a possibility.

The introduction to previous research on the molecular mechanisms governing the lack of ACKR3-G protein coupling was not intended to be all encompassing, but rather to highlight previous efforts to elucidate this process and justify our study of the role of dynamics. Due to the positions of the probes, we can only comment on the impact on TM6 movements and not other conformational changes. The steric clash reported in Yen et al. was in ICL2 and not directly tested here, so our observations do not preclude changes occurring in this region. We also do not claim that the active-like state resolved in our previous structures matches any specific state isolated here by smFRET.

(6) Line 83-85: "Having excluded other mechanisms we therefore surmised that the inability of ACKR3 to activate G proteins may be due to differences in receptor dynamics."Line 400-402: "It is possible that the active receptor conformation clashes sterically with the G protein as suggested by docking of G proteins to structures of ACKR3."

As mentioned above, we suspect the mechanisms governing the inability of ACKR3 to couple to G proteins may be more complex than one particular feature but instead due to a combination of several factors. Accordingly, we have not completely eliminated a contribution of steric hindrance as we described in Yen et al. Sci Adv 2022 and instead include it as a possibility. Following the line highlighted here, we list several alternatives:

“Alternatively, the receptor dynamics and conformational transitions revealed here may prevent formation of productive contacts between ACKR3 and G protein that are required for coupling, even though G proteins appear to constitutively associate with the receptor.”

And, at the end of the paragraph, we have added the following sentence:

“The atypical activation of ACKR3 does not appear to be dependent on any singular receptor feature and is likely a combination of several factors.”

(7) If the authors believe that the various ligands/mutations are only altering the distribution/dynamics of the same 3/4 conformations of CXCR4/ACKR3, respectively, is there a reason each FRET efficiency histogram is fit independently instead of constraining the individual components to Gaussian components with the same centroids, and/or globally fitting all datasets for the same receptor?

We performed global analysis across all data sets for each sample and condition. Since the peak positions of the various FRET states recovered in this way were consistent across treatments (Fig. S4,S6), we did not feel it was necessary to perform a further global analysis across all samples for a given receptor.

**Reviewer #3 (Recommendations For The Authors):**
The manuscript is well-written, the arguments are easy to follow and the figures are helpful and clear. Here are a few questions/suggestions that the authors might want to address before the paper will be published:(1) Include a table with kinetic rates between states in SI and have a brief discussion in the main text to support the trends observed in transition probabilities.

As noted above, determining rate constants for each of the state-to-state transitions will require a much larger set of experimental smFRET data than is currently available and will be the subject of future studies.

(2) The argument of state similarity (Figure S4 and S6)... why are the profiles not Gaussian, like in the fits on Figures S3 and S5, repectively? I would also suggest that once the number of states is chosen to do a global fit, where the FRET values of a certain sub-state across different conditions for one receptor are shared.

The state distributions presented in Figs. S4 and S6 (as well as throughout the rest of the paper) are derived from HMM fitting of the time traces themselves, and are not constrained to be Gaussian, whereas the GMM analysis in Figs. S3 and S5 are Gaussian fits to the final apparent FRET efficiency histograms.

Similar to our response to Review 2 above, due to the consistency of the fitted peak positions obtained across different conditions for a given sample, we did not feel that further global analysis was necessary.

(3) It is shown FRET changes from ~0.85 in the inactive (closed) state to ~0.25 in the active (open) state. How do these values match the expectations based on crystal structure and dye properties?

As noted in our response to Reviewer 1, translating the apparent FRET values using the assumed Förster distances for A555/Cy5 (per FPbase) suggest a change in D-A distance of ~30 angstroms, whereas the expected change from structures is ~16 Å. We suspect this discrepancy is due to the lipids immediately adjacent to the fluorophores, which may lead to the probes being constrained in an extended position when TM6 moves outwards, thus also reporting the linker length in the distance change. Additionally, such interactions may constrain the donor and acceptor in unfavorable orientations for energy transfer, which would also reduce the FRET efficiency in the active state. Since the calculated D-A distance changes appear too large for GPCR activation, we have opted to not make any structural interpretations. Instead, all of our conclusions are based on resolving individual conformational states and quantifying their relative populations, which is based directly on the measured FRET efficiency distributions, not computed distances.

(4) The results on the effect of CXCL12-P2G on CXCR4 are confusing...despite being an antagonist, this ligand stabilizes the "active state"...I am not sure if the explanation offered is sufficient that the opening of the intracellular cleft is not sufficient to drive the G protein coupling/activation.

We agree that the explanation related to the opening of the intracellular cleft being insufficient to drive G protein coupling/activation is speculative and we have removed that text. We now simply propose that the CXCL12 variants inhibit coupling of G proteins to CXCR4 or disrupt interactions necessary for signaling, as stated in the following text to the results on Pg. 8:

“Despite the ability of CXCL12_P2G_ and CXCL12_LRHQ_ to stabilize the active R* conformation of CXCR4, both variants are known to act as antagonists (20). This suggests that the CXCL12 mutants inhibit CXCR4 coupling to G proteins not by suppressing the active receptor population but rather by increasing the dynamics of the receptor state-to-state transitions. Our results suggest that the helical movements considered classic signatures of the active state may not be sufficient for CXCR4 to engage productively with G proteins.”